# Cryo-EM structures provide insight into how *E. coli* F$_1$F$_o$ ATP synthase accommodates symmetry mismatch

Meghna Sobti[1,2], James L. Walshe [1], Di Wu[3], Robert Ishmukhametov[4], Yi C. Zeng[1], Carol V. Robinson [3], Richard M. Berry[4] & Alastair G. Stewart [1,2]✉

F$_1$F$_o$ ATP synthase functions as a biological rotary generator that makes a major contribution to cellular energy production. It comprises two molecular motors coupled together by a central and a peripheral stalk. Proton flow through the F$_o$ motor generates rotation of the central stalk, inducing conformational changes in the F$_1$ motor that catalyzes ATP production. Here we present nine cryo-EM structures of *E. coli* ATP synthase to 3.1–3.4 Å resolution, in four discrete rotational sub-states, which provide a comprehensive structural model for this widely studied bacterial molecular machine. We observe torsional flexing of the entire complex and a rotational sub-step of F$_o$ associated with long-range conformational changes that indicates how this flexibility accommodates the mismatch between the 3- and 10-fold symmetries of the F$_1$ and F$_o$ motors. We also identify density likely corresponding to lipid molecules that may contribute to the rotor/stator interaction within the F$_o$ motor.

---

[1] Molecular, Structural and Computational Biology Division, The Victor Chang Cardiac Research Institute, Darlinghurst, NSW 2010, Australia. [2] Faculty of Medicine, St Vincent's Clinical School, UNSW Sydney, Kensington, NSW 2052, Australia. [3] Department of Chemistry, University of Oxford, Oxford OX1 3QZ, United Kingdom. [4] Clarendon Laboratory, Department of Physics, University of Oxford, Oxford OX1 3PU, United Kingdom. ✉email: a.stewart@victorchang.edu.au

A key component in the generation of cellular metabolic energy is the $F_1F_o$ ATP synthase, a biological rotary motor that converts the proton motive force (pmf) to adenosine tri-phosphate (ATP) in both oxidative phosphorylation and photophosphorylation[1–3]. The enzyme is comprised of two rotary motors, termed $F_1$ and $F_o$, that are coupled together by two stalks: a central "rotor" stalk and a peripheral "stator" stalk. The $F_o$ motor spans the membrane and converts the potential energy from the pmf into mechanical rotation of the central rotor that, in turn, drives conformational changes in the catalytic $F_1$ motor subunits to generate ATP from ADP and inorganic phosphate ($P_i$)[4,5]. However, in most ATP synthases there is a symmetry mismatch between the rotational steps made by the $F_1$ and $F_o$ motors—120° (corresponding to 3-fold symmetry of $\alpha_3\beta_3$ in $F_1$) and 36° (10-fold symmetry of the c-ring in $F_o$) in E. coli—that results in a non-integral $H^+$/ATP ratio[6]. To overcome this mismatch, the coupling between the two motors must be dynamic to enable the enzyme to function with high efficiency, with elastic energy stored within the complex facilitating different sized stepping between the $F_o$ and $F_1$ motors[7]. Hypotheses to account for this dynamic coupling have proposed flexibility within either the central[8,9] or peripheral stalks[10], and more recently the subunit that attaches the $F_1$ motor to the peripheral stalk[11] (termed δ in bacteria and OSCP in mitochondria), although the contribution made by each component has been controversial.

Although the $F_1F_o$ ATP synthase is found across most forms of life, the simplest form, that contains only eight different subunits, is present in E. coli and has been used extensively as a model system for ATP synthases[12]. Here we have used cryo-Electron Microscopy (cryo-EM) on detergent-solubilized cysteine free E. coli ATP synthase[13] to probe its structure and understand its rotational dynamics and coupling. We describe an ensemble of 3.1–3.4 Å resolution structures of the enzyme in a series of conformational sub-states that enable key functional features to be identified. The higher resolution information obtained in the $F_o$ region identifies lipids that may contribute to the rotor/stator interface and increase the interacting surface between the $F_o$ stator and rotor ring. In addition to generating a comprehensive structural model of E. coli $F_1F_o$ ATP synthase that provides a framework to interpret mutagenesis studies, we also describe torsional flexing of the complex and a rotational sub-step of the $F_o$ motor c-ring associated with long-range conformational changes. These data indicate a model of how elastic coupling between the $F_1$ and $F_o$ motors is mediated by a dynamic, flexible peripheral stalk.

## Results

**Rotational sub-states of E. coli $F_1F_o$ ATP synthase.** Cryo-EM maps of cysteine free E. coli $F_1F_o$ ATP synthase in the presence of 10 mM MgADP were obtained at 300 kV using methods similar to those in previous studies[14,15] (Fig. 1 and Supplementary Figs. 1 and 2). MgADP was used in an attempt to lock the rotor of the $F_1$-ATPase in a single rotational position[16], to investigate the flexible coupling between the $F_1$ and $F_o$ motors, as well as the contribution made by nucleotides on the regulation of the $F_1$ motor. Hence, the $F_1F_o$ ATP synthase imaged here should not be undergoing ATP synthesis or hydrolysis, and therefore should not be rotating under the conditions imaged in this study. These maps generated far superior structural information than observed previously for this complex[14,15], with the overall resolution improving from ~5 Å in previous studies to ~3 Å in this study, allowing most of the sidechains to be assigned in the model. Furthermore, masked sub-classifications focused on the $F_o$ stator (subunit a and the N-termini of the b subunits) identified a series of intermediates that provided detailed information on

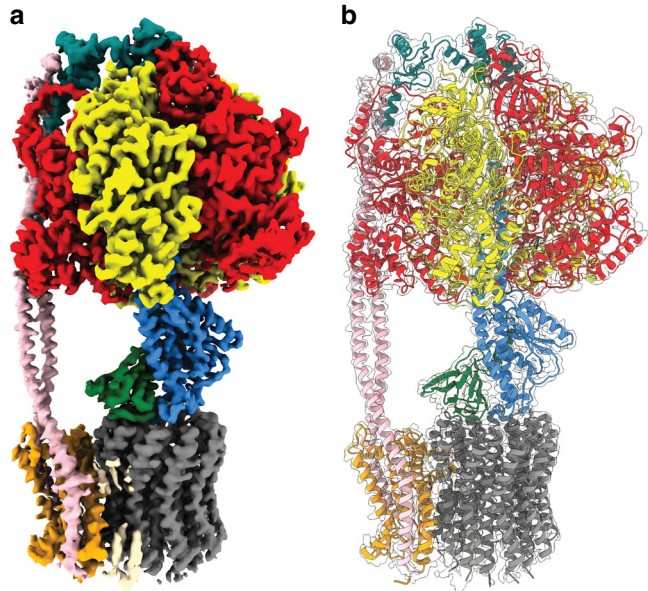

**Fig. 1 Cryo-EM map and atomic model of E. coli $F_1F_o$ ATP synthase.** Subunit α in red, β in yellow, δ in teal, ε in green, γ in blue, a in orange, b in pink, c in gray and potential lipids in wheat. Very weak density likely corresponding to detergent was removed for clarity. **a** Cryo-EM map shown as surface and (**b**) atomic model shown in transparent cryo-EM map. Sub-state 3A is shown.

conformational sub-states accessible to the complex (Supplementary Figs. 2–4).

The maps obtained identified a series of states that correspond to different rotational steps of E. coli $F_1F_o$ ATP synthase. The three major rotational states identified previously (termed States: "State 1", "State 2", and "State 3", assigned to describe the enzyme operating in an ATP hydrolysis direction), in which the central stalk is rotated by ~120° relative to peripheral stalk[14], were also obtained in this study with resolutions of 2.9, 3.1, and 3.0 Å, respectively (Supplementary Fig. 2). Although these reconstructions gave the highest numerical resolution, the Fourier Shell Correlation (FSC) appeared to be dominated by the $F_1$-ATPase, with information in the $F_o$ region blurred significantly (Supplementary Figs. 3 and 4). Hence, masked classification focused on the $F_o$ stator was used on each of the main states to reveal sub-states describing movements of the complex along with increased detail in the $F_o$ region (Supplementary Figs. 2–5). By examining the position of subunit γ in each of the sub-states, we were able to unambiguously assign the c subunits, so that their relative position could be compared between each sub-state (Supplementary Fig. 6). Focused classification of rotational State 1 highlighted five sub-classes, here termed Sub-states 1A, 1B, 1C, 1D, and 1E; with resolutions of 3.1, 3.3, 3.1, 3.2, and 3.3 Å, respectively (Supplementary Fig. 2). The defining theme of these sub-states was flexing of the central and peripheral stalks, with the majority of the flexibility seen in the peripheral stalk which bends and twists, together with minor movements within the central stalk (Fig. 2, Supplementary Figs. 7 and 8 and Supplementary Movie 1). Independently, the movements seen between Sub-states 1B-E do not facilitate rotation of the $c_{10}$ ring and instead illustrate torsional flexing of the entire complex. However, comparing Sub-state 1A with 1B-E shows a ~36° rotation of subunits $\alpha_3\beta_3\gamma\epsilon c_{10}$ (corresponding to the $F_1$ motor plus the c-ring) relative to the $F_o$ stator (Fig. 2 and Supplementary Movie 2). Importantly, this rotation describes a ~36° movement of the c-ring in the $F_o$ motor without any corresponding rotation within the $F_1$ motor (Supplementary Fig. 9): the $\alpha_3\beta_3\gamma\epsilon c_{10}$ subunits are rotating as

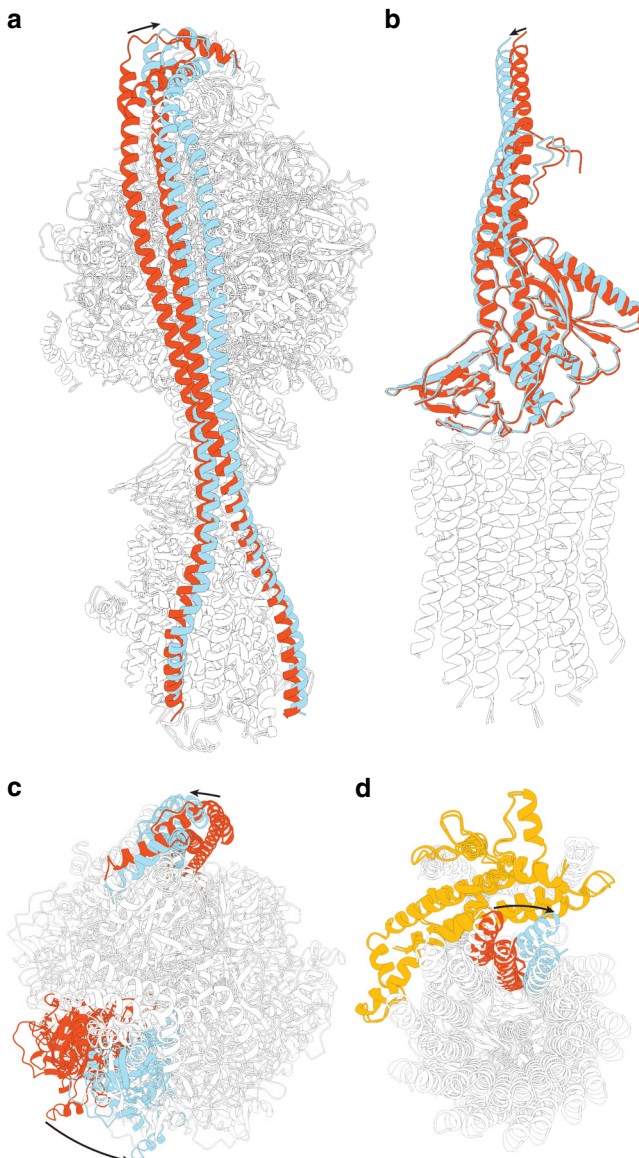

**Fig. 2 The peripheral stalk bends and twists to facilitate c-ring rotation. a** Molecular models of Sub-states 1A and 1E (see Supplementary Fig. 2 for details) superposed on stator subunit a. The peripheral stalk, colored in red in Sub-state 1A and cyan in Sub-state 1E, twists and bends to facilitate $F_o$ sub-stepping. **b** Molecular models of the central rotor of Sub-states 1A and 1E superposed on the c-ring highlight a small movement at the interface between the stalk and c-ring, which facilitates $F_o$ sub-stepping. **c** Superposition on stator subunit a as viewed from above highlights rotational movement of the $\alpha_3\beta_3\gamma\varepsilon c_{10}$ relative to the stator (black arrows), with the same $\alpha$ subunit colored red for Sub-state 1A and cyan for Sub-state 1E. **d** Superposition on stator subunit a as viewed from below shows the ~36° rotation of $\alpha_3\beta_3\gamma\varepsilon c_{10}$ relative to the a subunit (colored in orange), with the same c subunit labeled red for Sub-state 1A and cyan for Sub-state 1E.

a rigid body relative to the $F_o$ stator, with the movement mediated mainly by changes in the peripheral stalk. The focused classifications of States 2 and 3 also revealed sub-states, however these showed only torsional flexing similar to that seen in Sub-states 1 B-E, and hence no movement of the c-ring was observed within sub-states of States 2 and 3.

Thus, across our maps of *E. coli* ATP synthase, the rotor (consisting of the subunits $\gamma\varepsilon c_{10}$) was observed in four discrete

rotational positions relative to the $F_o$ stator: one in each of States 2 and 3, and two in State 1 (Fig. 3a and Supplementary Movie 2). Sub-state 1A was in one rotational sub-state (here termed Sub-state 1'), whereas Sub-states 1B-E were rotated by a single c subunit relative to Sub-state 1' (here termed Sub-state 1"). We were unable to confidently classify the remaining ~40% of the particles into discrete rotational sub-states due to the weak peripheral stalk density in maps generated with them. It is our assumption that these unclassified particles represent molecules with conformations that are around the states identified and, because of their torsional variability, have insufficient signal for the sorting algorithm to succeed in classifying them. Though we cannot rule out that these particles represent conformations well beyond those identified in this study. Across all the sub-states that we are able to identify confidently, the general arrangement of the stator subunit a and rotor c-ring in $F_o$ remained essentially the same, preserving a similar interacting surface between the rotational states (Supplementary Fig. 6). Furthermore, the position of the $\gamma$ subunit relative to the $\alpha$ and $\beta$ subunits was in the same rotational position in all sub-states (Supplementary Fig. 9), although rotated relative to the peripheral stalk, with the $F_1$ enzyme locked in the same rotational state across all structures observed in this study.

**The $F_o$ motor stator/rotor interaction.** The maps generated after classification into sub-states provided substantially improved detail in the $F_o$ region, compared to that seen without masked classification or that seen previously[14,15], with sidechain density observed for the majority of residues (Supplementary Figs. 4 and 5). However, the reconstructions were still anisotropic due to preferential alignment to the $F_1$-ATPase, so focused refinement using a mask of the $F_o$ region was performed on Sub-state 3A resulting in a detailed map to 3.3 Å resolution of the $F_o$ motor (Fig. 4, Supplementary Figs. 2 and 10). This map enabled a detailed model of the membrane-embedded a, b and c subunits to be built (Fig. 4b), which was fitted and refined to all other sub-states. The model generated complements those that have been determined for other species, particularly for the related *Bacillus sp.* PS3[17] and chloroplast enzymes[18]. Importantly, because *E. coli* $F_1F_o$ ATP synthase has been a model system for studying ATP synthase for decades[12], the model described here provides a detailed structural framework to interpret the wealth of mutagenesis studies that have been performed on this enzyme. A particularly interesting feature seen in this study are the positions of aArg210 and cAsp61, residues that are known to be essential for proton translocation[19]. Density for these residues was resolved particularly well and suggests them to be in close proximity to each other (Fig. 4b). Furthermore residues aAsn214, aHis245, aAsn119, and aGlu219, that are also known to be important from mutagenesis studies[20], form a chain of residues between the aqueous channel and rotation path of cAsp61 (Supplementary Fig. 11).

As a coordinated metal ion has been proposed to mediate protonation of the c subunit in related ATP synthases[11,18,21], we closely inspected our maps for ion-like density in this region. We failed to find any ion-like density adjacent to residues aGlu219 and aIle223, residues equivalent to those shown to coordinate an ion in *Polytomella* mt. ATP synthase (Supplementary Fig. 12). However, we did observe a weak spherical density in the rotation path of cAsp61, adjacent to aAsn214 and aHis245, which are essential for function in *E. coli* $F_1F_o$ ATP synthase (Fig. 4 and Supplementary Fig. 11). Given the location and shape of this density, it could represent an ion or water molecule at the protonation site of c-ring. As the distances to neighboring polar atoms are too large to hydrogen bond and the cavity in which this density resides has a van der Waals radius ~ 3 Å (Supplementary Fig. 12), the identity of this density is unlikely to be a single water

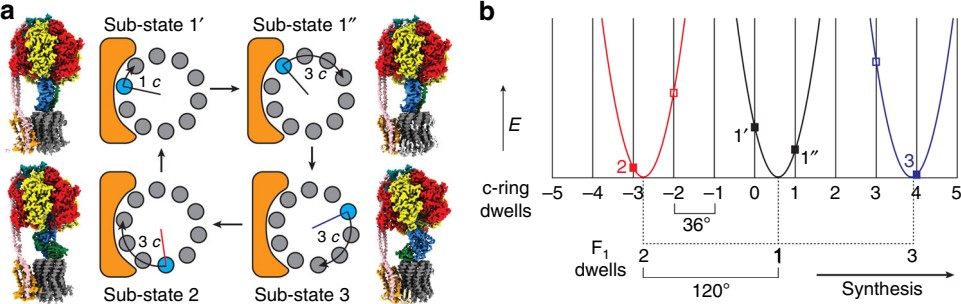

**Fig. 3 The F_o motor is observed in four discrete rotational positions. a** Simple schematic to describe c-ring positions observed in this study. Arrows show rotation of the c-ring between each step in the ATP synthesis direction (same c subunit colored in cyan and all others in gray) with colored radii (black in Sub-state 1′ and 1″, blue in Sub-state 3 and red in Sub-state 2) describing relative position to the subunit a colored orange (see Supplementary Fig. 6 for assignment of the rotational position of the c-ring). Each schematic is accompanied by a cryo-EM map representative for the state discussed. Sub-state 1′: Sub-state 1A, Sub-state 1″: Sub-state 1C, Sub-state 3: Sub-state 3A and Sub-state 2: Sub-state 2B. **b** Simple schematic to describe the stored elastic energy (E) versus F_o rotation angle, assuming a linear (Hookean) spring with a zero-point that rotates 1/3 of a revolution between each of F_1 states 1-3. The parabolic curves represent the energy stored in this spring in each F_1 state, as a function of the rotation angle of F_o: $E = \frac{1}{2}\kappa x^2$, where $\kappa$ is the spring constant and $x$ the displacement of the spring from its minimum. The minima for the three F_1 states were assumed to be equally spaced, 120° apart, and the stiffness of the spring was assumed to be the same in all three states. The angles of these minima relative to the 10 F_o dwells (vertical lines) were adjusted by hand to set the energy of Sub-state 1″ lower than that of Sub-state 1′ (as required if the 3-fold higher count of particles classified as Sub-state 1″ relative to Sub-state 1′ is representative of the equilibrium occupancy of these states), while at the same time setting the energies of Sub-states 2 and 3 close to their minima. The four observed sub-states are those with elastic energy low enough to be accessed thermally, shown as black filled squares. Sub-states that were not observed in this study and cannot be accessed thermally under these conditions are shown as empty boxes. F_1 dwells State 2 (red) and State 3 (blue) align close the c-ring dwells and therefore we were only able to observe F_1F_o in single c-ring positions. F_1 dwell State 1 (black) aligns mid-way between two c-ring dwells, so we were able to observe F_1F_o in two separate c-ring positions.

molecule surrounded by vacuum. Instead this region could correspond to a cluster of water molecules or a hydrated ion such as Magnesium, Sodium, or Phosphate. However, assigning such a density at ~3 Å resolution is challenging and clearly further work would be required to evaluate the identity and importance of this region. Hence the deposited co-ordinates do not contain any atoms in this space.

The peripheral stalk of *E. coli* ATP synthase is constructed by a homodimer containing two b subunits. Each subunit consists of a long alpha helix with three sections (Supplementary Fig. 14). The N-terminal section (b1-b45) resides in the membrane and braces against the a subunit, the middle section (b46-b135) forms a ~130 Å long right-handed coiled coil and the C-terminal section (b136–b154) loops back to cap the attachment to the F_1-ATPase. The helices within the parallel right-handed coiled coil are offset by 5½ residues with respect to one another. This offset shows a striking correlation to the arrangement that was predicted previously using crosslinking studies[22], showing a staggered homodimeric right-handed coiled coil (Supplementary Fig. 15). Although the first four residues of subunit a were not resolved in our maps, residues 8 to 16 form a helix that sits on the periplasmic side of the membrane and which interacts with the N-terminus of the adjacent b subunit and the C-terminus of one of the c subunits (Supplementary Fig. 16a). Residues 42–88, 100–107 and 147–156 of subunit a also appear to reinforce the interactions with the other b subunit (Supplementary Fig. 16b).

When all the density corresponding to protein had been assigned, tube-like non-protein density likely corresponding to lipid molecules were observed in multiple locations within the membrane bound F_o-motor (Supplementary Fig. 17). The strongest of these densities were observed between residues aI225–aQ234 and a single c subunit, where three of these tubes of density were present on the periplasmic side and one on the cytoplasmic side of the enzyme (Fig. 5 and Supplementary Movie 3). This additional density was observed in most of the maps but was clearest in the highest resolution maps, Sub-states 1C and 3A, and the focused F_o map of Sub-state 3A. Without the

lipid(s), the interaction surface between subunit a and the c-ring would be limited to three of the ten c subunits, whereas with the lipid-mediated interaction this is increased by an additional c subunit to four (Fig. 5). Although the detail of the maps was insufficient to establish the identity of the lipid(s) in this bridge region, the arrangement of these lipid(s) suggests an intriguing mechanism whereby, in addition to direct protein-protein contacts, the strength of the stator/rotor interaction is increased by using lipid(s) to bridge between the interacting surfaces. As well as the strong density observed for the lipid bridge, weaker density corresponding to the lipid bilayer was also observed around the F_o motor, with 61 lipid chain-like densities (Supplementary Fig. 17), as well as the lipid plug in the center of the c-ring (Fig. 5b). These lipids encompassed the entire c-ring and are reminiscent to that seen around the V_o motor from *Saccharomyces cerevisiae*[23]. To investigate whether the densities observed could be attributed to lipids, LC-MS-based lipidomics[24] was performed on the same detergent-solubilized *E. coli* F_1F_o ATP synthase imaged in this study. Phosphatidylethanolamines, phosphatidylglycerols and cardiolipins were all observed, with an increase in the relative abundance of cardiolipins compared to *E. coli* membrane (Supplementary Fig. 18), showing that *E. coli* lipids were co-purified with the protein.

**MgADP induces a conformational change in the F_1 motor.** The maps presented here were generated from material imaged in the presence of 10 mM MgADP and all showed the enzyme in the "autoinhibited" state, with the C-terminal domain of the ε subunit (εCTD) in the "up" inhibited position interacting with the F_1 catalytic subunits (Fig. 6a). The conformation of the F_1 motor is different to that seen in previous cryo-EM studies[14,15], in that the catalytic "β1" subunit (as defined in reference 25) has bound ADP, Mg and P_i and is seen in a "half-closed" state, which is prevented from closing fully by the εCTD (Fig. 6a, Supplementary Figs. 19 and 20). This conformation is remarkably similar to that seen in the crystal structure of the isolated F_1 motor[25]

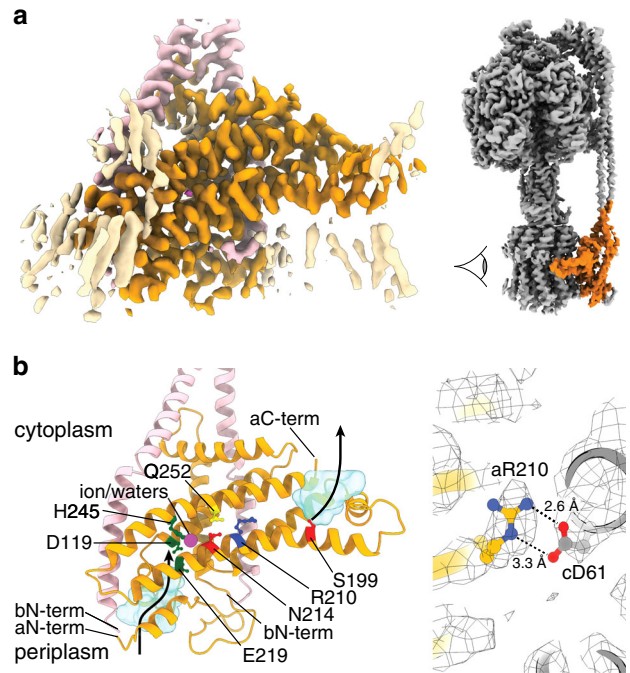

**Fig. 4 The $F_o$ stator of *E. coli* $F_1F_o$ ATP synthase. a** Sub-state 3A $F_o$ focused cryo-EM map of the *E. coli* $F_o$ stator, masked to remove c-ring for clarity (subunit a in orange, subunit b in pink and lipid-like densities in wheat). Highlighted orange region in the intact enzyme on right shows area depicted, with viewpoint indicated with an eye. **b** Molecular model of the $F_o$ stator, as viewed from the c-ring, showing discussed residues at the stator rotor interface. Colors as in Fig. 1 (subunit a in orange and subunit b in pink) with the proton half-channels shown as light blue inverted surfaces (made with the program HOLLOW[60]) and potential proton path shown with black arrow. Essential aArg210[27] (dark blue) is located in the center of the membrane, adjacent to cAsp61[28] (shown in c). aGln252 is situated proximally to aArg210 and function can be retained when these residues are interchanged with one another[37]. aGlu219[32], aAsp119, aHis245[33], and aAsn214, which are required for proton translocation, are situated between the end of the periplasmic half-channel and cAsp61, adjacent to density that could be a ion/water molecule (magenta sphere). aSer199 is situated at the beginning of the cytoplasmic proton channel. N-termini and C-termini of subunits a and b labeled.

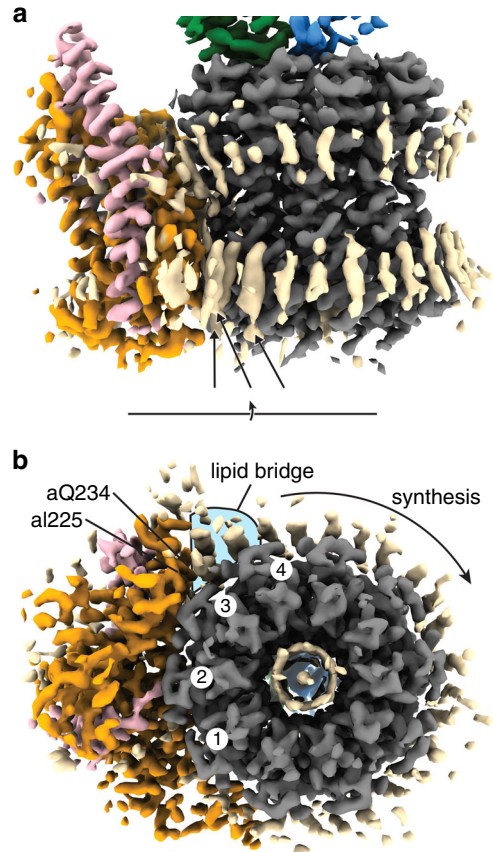

**Fig. 5 Lipid bridges contribute to the stator/rotor interaction.** Cryo-EM density of Sub-state 3 A, centered on the $F_o$ region. Density corresponding to lipids (colored in wheat) "bridge" the interaction between subunits a (orange) and c (gray). **a** When viewed from within the membrane, density can be seen for lipids around the $F_o$ motor. Three lipid-like densities on the periplasmic side (labeled with black arrows) are the strongest of these densities, suggesting that they are well ordered. **b** When viewed from the periplasmic side (view rotation axis shown), the interaction between residues Ile225–Gln234 of the stator subunit a and c ring is facilitated by three lipid densities (highlighted with blue background), increasing the interaction to four rather than three c subunits (numbered one white circles).

(Supplementary Fig. 19). However, inspection of the nucleotide binding pockets in the cryo-EM structures showed density for nucleotide in all six of $\alpha$ and $\beta$ subunits, whereas only four were observed to be occupied in the isolated $F_1$ motor crystal structure[25]. In the cryo-EM maps of the intact $F_1F_o$ enzyme in the presence of 10 mM MgADP, all $\alpha$ subunits contained ATP and $Mg^{2+}$, whereas the contents of the $\beta$ subunits varied: $\beta 1$ bound ADP, $Mg^{2+}$ and $P_i$, $\beta 2$ bound only ADP, and $\beta 3$ bound ADP and $Mg^{2+}$ (Fig. 6b and Supplementary Fig. 20). This composition contrasted with that observed in the crystal structure of *E. coli* $F_1$-ATPase soaked in 1 mM AMP-PNP, where $\beta 1$ bound ADP, $Mg^{2+}$ and $SO_4^{2-}$, and $\beta 2$ and $\beta 3$ bound only $SO_4^{2-}$[25]. The presence of nucleotide in all $\alpha$ and $\beta$ subunits in our cryo-EM maps, likely reflects the relatively high concentration of MgADP used during sample preparation.

## Discussion

The high-resolution information obtained by cryo-EM analysis of *E. coli* ATP synthase presented here, provides information on how the complex adapts its conformation during rotation to accommodate symmetry mismatch between the $F_o$ and $F_1$ components, the potential role of lipids to increase the rotor/stator interface, and the conformational changes that are induced by binding of MgADP.

Strikingly, of the three $F_1$ rotational states, defined by the angle of the $\gamma$-subunit relative to the peripheral stalk, only one showed two $F_o$ rotational sub-states, as defined by the rotation angle of the $c_{10}$-ring relative to the stator. This can be explained as a consequence of the potential elastic flexibility in the peripheral stalk ("the spring") seen in this study and the symmetry mismatch between the $F_1$ and $F_o$ motors. In State 1 the un-stretched spring corresponds to an $F_o$ rotation angle in-between two of its 10 preferred rotational sub-states (Fig. 3b). These two correspond to the observed Sub-states 1 and 1", which have similar quantities of stretch (close to 1/20 of a revolutions), and therefore also energy stored in the spring. By contrast, in States 2 and 3 the spring is close to its relaxed energy state in the observed single $F_o$ rotary states, and the adjacent $F_o$ rotary states are inaccessible

**a**

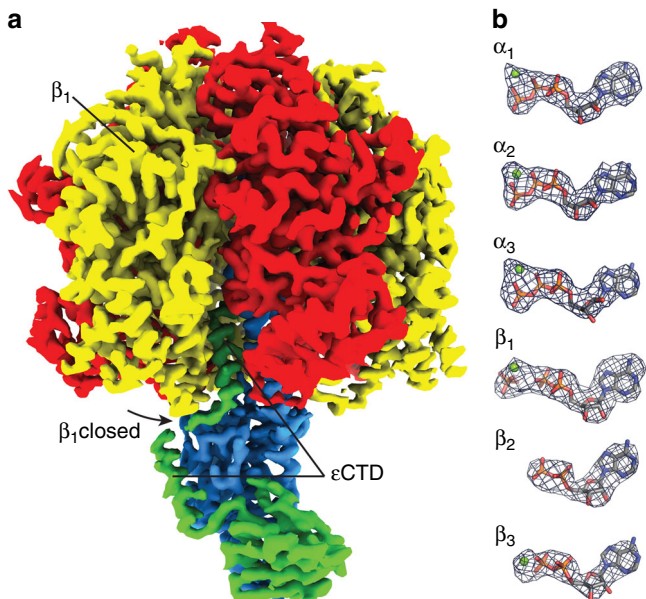

**b**

α₁

α₂

α₃

β₁

β₂

β₃

**Fig. 6 The F₁ motor changes conformation to the autoinhibited state in 10 mM MgADP. a** Masked maps of the α, β, γ, and ε subunits showing the β₁ subunit (labeled) "half-closed" (black arrow describes direction of movement in β₁ subunit) onto the εCTD (labeled). The C-terminal domain of the ε subunit is in an "up" conformation. **b** Nucleotide and Mg²⁺ occupancy of the α and β subunits, with difference density (blue mesh) of nucleotides shown (with equivalent Mitochondrial F₁ nomenclature[4]: β₁ = β_DP, β₂ = β_E, β₃ = β_TP[25]).

because thermal excitation is insufficient to stretch the spring through the required 1/10 of a revolution (Fig. 3b). Hence, the flexibility of the peripheral stalk is sufficient to provide the conformational freedom to accommodate the symmetry mismatch. However, as ~40% of the particles processed in this study were not classified into the sub-states observed, it is certainly speculative to infer the elastic energy stored in this system. Hence further studies on the exact nature of this coupling will be needed to understand the interaction fully. Still, the simple observation of four discrete rotational sub-steps in the F_o motor highlights how a symmetry mismatch between the 3-fold F₁-ATPase and 10-fold F_o motor can be accommodated (Fig. 3a).

In the structures we obtained for the *E. coli* enzyme we did not observe any significant hinge movement in the δ subunit (Supplementary Fig. 21), that corresponds to OSCP in mitochondrial ATP synthases, and which has been proposed to provide flexible coupling in other systems[11]. In contrast to *Polytomella* mt. ATP synthase, where the peripheral stalk remains essentially unchanged during rotation[11], the *E. coli* peripheral stalk bends and twists substantially. This difference could derive from *Polytomella* mt. ATP synthase having a large bridged peripheral stalk, the structure of which appears likely to inhibit substantial movement within this component[11] so that, in this species, flexibility is generated primarily by the junction between it and the F₁ motor that is formed by the OSCP subunit. However, it has been proposed[11] that this hinge is also found in bovine mitochondrial ATP synthase[26], raising the possibility that the mechanism by which the stator flexes might be different between eukaryotes and prokaryotes. Although further work will be needed to investigate this proposal, it raises the possibility that the hinges present in the *E. coli* stator could represent a novel drug target, especially since a number of agents are known to target the flexing OSCP subunit in mitochondrial ATP synthases[11]. An alternative hypothesis is

that the two mechanisms; twisting and flexing in the peripheral and central stalks (seen in this study on *E. coli*), and a hinge in subunit δ/OSCP (seen in *Polytomella*[10]), could combine to allow a fluid motion across a ~60–80° sub-step of the enzyme. The data presented in this manuscript shows the enzyme in its auto-inhibited conformation, with the F₁-ATPase in a single rotational state. Hence, further work will need to be performed on this enzyme to investigate whether the peripheral and central stalks retain the same properties when the enzyme is under load either synthesizing or hydrolyzing ATP.

The high-resolution information obtained for the intact *E. coli* complex provides a framework to further understand the proton path in F_o, along with the multitude of mutagenesis studies that have been performed in this region (Fig. 4b). For example, density for aArg210[27], which is essential for F_o rotation, is positioned adjacent to the proton translocating residue cAsp61[28] on the rotor ring (Fig. 4b). Radiation damage in cryo-EM causes the preferential loss of carboxylates, making the confident assignment and precise building of glutamates and aspartates challenging[29]. Other studies of ATP synthases have focused on organisms that utilize a glutamic acid to translocate protons, whereas *E. coli* F₁F_o ATP synthase utilizes aspartic acid, cAsp61, to translocate protons. As aspartic acid contains a shorter sidechain than the glutamic acid, the possible rotamer positions after electron irradiation are fewer giving an advantage over other systems when attempting to assign the conformation of this sidechain. Although we cannot unequivocally assign the precise nature of this interaction, the cAsp61 adjacent to aArg210 appears to be in an outward facing conformation compared to cAsp61 residues exposed to the membrane (Fig. 4b), which corroborates crystallographic studies on the isolated c-ring[30]. The position of aArg210 likely prevents short circuiting of the protons half-channels, as suggested by other studies[19]. Unlike other structural studies on related ATP synthases[11,18,31], the interface between the a and c subunits remained largely similar across the classified maps. This could be a hallmark of the εCTD/MgADP inhibited state, a specific feature of *E. coli* F₁F_o ATP synthase or an artifact of the methods used to classify the particles. Further studies on uninhibited enzymes and ATP synthases from other organisms will be needed to delineate this structural feature.

A deep funnel-like evagination can be observed on the cytoplasmic side of the membrane that likely facilitates deprotonation of the c-ring near aSer199, which is adjacent to cAsp61. The proton path on the periplasmic side of the membrane less direct, with a smaller channel leading to aGlu219 (Fig. 4b). Residues aGlu219[32], aAsp119[20], aHis245[33], and aAsn214[34], which have been found to be required for proton pumping[20,35], form a chain of residues between the periplasmic channel and the rotation path of cAsp61 (Supplementary Fig. 11). Moreover, mutagenesis studies on *E. coli* F₁F_o ATP synthase have shown that residues aGlu219 and aHis245 can be substituted for one another and still retain function[36], suggesting that their function is coupled. Although a coordinated metal ion has been proposed to mediate protonation of the c subunit[11] in related ATP synthases via this route[11,18,21], close inspection of our maps (Supplementary Fig. 13) failed to show any non-peptide density in this region. This could be explained by the fact that the residues which coordinate the ion in *Polytomella* mt. ATP synthase, aHis248 and aHis252, correspond to aGlu219 and aIle223 in the *E. coli* enzyme and so the local geometry needed for metal binding would be lost in the bacterium. Instead we observed weak ion/water-like density at the rotor/stator interface adjacent to the functionally essential residues aAsn214 and aHis245, which would directly interact with the proton binding cAsp61 residue as the c-ring rotates potentially facilitating protonation (Fig. 4b and Supplementary

Fig. 11). However, the observation of such a ion/water density is highly speculative and future more detailed maps along with other experiments will be needed to confirm the presence and significance of such an entity.

Our model also places aGln252 adjacent to aArg210 (Fig. 4b), and multiple studies have shown that if these residues are substituted for one another, function can be retained[37,38] with the arginine residue still being able to contact cAsp61 and facilitate rotation in our model. The maps also show density for residues 10-20 of subunit a, which is located adjacent to the N-terminus of subunit b (Supplementary Fig. 16a). Truncation of these amino acids has been shown to interfere with the interaction between the a and b subunits, resulting in a non-functional enzyme[39]. This, together with multiple contacts between both b subunits and transmembrane helices of subunit a (Supplementary Fig. 16b), point to this being a load-bearing element that clamps subunits b together and prevents them from deforming during rotation. Mutations in this area have a detrimental effect on the enzyme and result in changes in crosslinking analyses[39,40].

The lipid bridge we observe at the rotor/stator interface (Fig. 5) suggests a mechanism whereby lipids can form a functional component of the complex to facilitate transient interactions between the stator and rotor. Although lipids have been identified elsewhere in previous studies[11,41], the lipids we observe in $E.\ coli$ $F_1F_o$ ATP synthase appear to mediate an interaction between the rotor and stator. This lipid bridge increases the interacting surface between subunit a and the c-ring so that it encompasses four of the ten c subunits, with three interacting directly through protein-protein contacts and one through the lipid bridge (Fig. 5). Although it is unclear why the $E.\ coli$ enzyme employs such a mechanism, an attractive hypothesis would be that the lipids facilitate a transient interaction that would decrease the energy required for rotation while still contributing to the interaction surface that is important to maintain complex integrity. $F_1F_o$ ATP synthases from other organisms with larger c-rings, such as chloroplast[18], do not appear to have the specific local geometry that would facilitate the lipid bridge we observe in $E.\ coli$, as the horizontal[42] or inclined membrane helices can bend to the curvature of the c-ring, suggesting that this feature could be restricted primarily to bacteria or ATP synthases with smaller c-ring sizes. For example, in chloroplast $F_1F_o$ ATP synthase[18] the a subunit is able to interact with four of the fourteen c subunits in the rotor ring and the long transmembrane helices in subunit a can adapt to the curvature of the c-ring. Inspection of the maps generated from porcine[43] and $Bacillus\ sp.$ PS3[17] $F_1F_o$ ATP synthase in light of our finding, shows similar densities in this region, though this was not highlighted by the authors of these studies (Supplementary Fig. 22), suggesting that this lipid mediated interaction may be conserved across some species containing smaller c-rings. During synthesis the $F_o$ motor would rotate in a clockwise direction when viewed from the periplasm (Fig. 5). This would result in c-ring rotation away from the lipid bridge, hence if the lipid were to remain in place under synthesis the interaction to the a subunit must be stronger than that to the c subunit. The role lipids play in the function of membrane proteins has been an area of considerable interest and controversy, with the lipid environment often appearing to influence function. Mitochondrial ATP synthase requires cardiolipin to function, which has been shown to bind to conserved lysine residues in metazoan ATP synthases[44]. Lipids functioning as bridges between protein subunits within and between complexes may facilitate transient dynamic interactions within membranes and several studies have identified lipids that are essential for the oligomerization of a considerable range of membrane proteins[45].

The cryo-EM structure of the $F_1$ motor described here also provided information on the structural changes and nucleotide occupancy introduced by the binding of MgADP. Currently, only two studies have provided cryo-EM maps of an ATP synthase in which conditions have been manipulated to image the sample with added nucleotide[15,46]. One of these studies, which exposed the $E.\ coli$ enzyme to 10 mM MgATP for 45 s[15], described the molecular events that occur upon ATP binding and showed that the C-terminal domain of the inhibitory subunit $\varepsilon$[47] was removed from the central cavity when ATP was bound. Our present maps show that a short incubation of 10 mM MgADP induces the catalytic subunits to bind this ADP (Fig. 6) which, in turn, induces a partial "closure" of the $\beta$ subunit that is blocked in a "half-closed" position by the inhibitory $\varepsilon$ subunit in a manner reminiscent of that seen in the crystal structure of isolated $E.\ coli$ $F_1$-ATPase[25] (Supplementary Fig. 19). Interestingly, the nucleotide occupancy differed to that in the isolated $F_1$ enzyme, with ADP seen in all catalytic sites and $P_i$ bound along with MgADP in the "half-closed" $\beta$ subunit (Supplementary Fig. 20). As no $P_i$ was intentionally introduced into the system, it is likely that this component was either introduced with the purchased ADP as a contaminant or is already present in the enzyme preparation. Trypsin treatment of $E.\ coli$ $F_1F_o$ ATP synthase after incubation with 5 mM MgADP+$P_i$[48] has previously suggested that the $\varepsilon$CTD would adopt a down confirmation in the presence of MgADP+$P_i$ and single molecule studies[49,50] have also suggested that the conformation of the $\varepsilon$CTD would differ in the presence of MgADP+$P_i$. However, our cryo-EM work did not detect any structures in which the $\varepsilon$ subunit was in a down confirmation, which could be due to the concentration of $P_i$ being below the threshold to induce changes in the $\varepsilon$ subunit or due to the short incubation time used. Nevertheless, the observation that MgADP induces a single $\beta$ subunit to close but is blocked half-way by the $\varepsilon$CTD in the up position, strongly suggests that a key function of the $\varepsilon$ subunit is to increase the efficiency of the enzyme by preventing $E.\ coli$ $F_1F_o$ ATP synthase from entering a low energy MgADP inhibited state, which is known to inactivate the enzyme[16,51] (Supplementary Movie 4).

In summary, the cryo-EM study presented here has generated a comprehensive structural model of $E.\ coli$ $F_1F_o$ ATP synthase, providing a framework to understand mutagenesis studies together with yielding insight into the flexibility of the peripheral and central stalks. The range of rotary sub-states together with the lipid bridge observed describe an attractive mechanism by which the $F_1$ and $F_o$ motors can be coupled with minimal energy loss. In addition, the structural rearrangement observed upon binding of MgADP suggests that the $\varepsilon$ subunit can function as a brace to prevent the complex falling into the MgADP inhibited state.

## Methods

**Protein purification.** The $E.\ coli$ $F_1F_o$ ATP synthase protein was prepared as described in Sobti et al. 2020[52]. Cysteine free $E.\ coli$ ATP synthase (plasmid generated in Ishmukhametov et al. 2005[13], which has had all cysteines residues substituted with alanine and a His-tag introduced on the $\beta$ subunit) was expressed in $E.\ coli$ DK8 strain[53]. Cells were grown at 37 °C in LB medium supplemented with 100 μg ml$^{-1}$ ampicillin for 5 h. The cells were harvested by centrifugation at 5000 × $g$, providing ~1.25 g cells per litre of culture. Cells were resuspended in lysis buffer containing 50 mM Tris/Cl pH 8.0, 100 mM NaCl, 5 mM MgCl$_2$, 0.1 mM EDTA, 2.5% glycerol and 1 μg ml$^{-1}$ DNase I, and processed with three freeze thaw cycles followed by one pass through a continuous flow cell disruptor at 20 kPSI. Cellular debris was removed by centrifuging at 7700 × $g$ for 15 mins, after which the membranes were collected by ultracentrifugation at 100,000 × $g$ for 1 h. The ATP synthase complex was extracted from membranes at 4 °C for 1 h by resuspending the pellet in extraction buffer consisting of 20 mM Tris/Cl, pH 8.0, 300 mM NaCl, 2 mM MgCl$_2$, 100 mM sucrose, 20 mM imidazole, 10% glycerol, 4 mM digitonin and EDTA-free protease inhibitor tablets (Roche). Insoluble material was removed by ultracentrifugation at 100,000 × $g$ for 30 min. The complex was then purified by binding on Talon resin (Clontech) and eluted in 150 mM imidazole, and further

purified with size exclusion chromatography on a 16/60 Superose 6 column equilibrated in a buffer containing 20 mM Tris/Cl pH 8.0, 100 mM NaCl, 1 mM digitonin and 2 mM MgCl₂. The purified protein was then concentrated to 11 μM (6 mg ml⁻¹), and snap frozen and stored for grid preparation. The protein used in this study came from the same preparation as used in Sobti et al. 2019[15], which contains activity assays for this enzyme, showing ~80 s⁻¹ turnover per enzyme (~9 μmoles min⁻¹ mg⁻¹) even after 8 h at room temperature, as well as sensitivity to DCCD[14] showing coupled $F_1F_o$ complex. SDS PAGE was performed to assess protein purity (Supplementary Fig. 23), with bands identified using mass spectrometry.

**Cryo-EM grid preparation**. One microlitre of 100 mM ADP/100 mM MgCl₂ (pH 8.0) was added to an aliquot of 9 μl of purified cysteine free *E. coli* $F_1F_o$ ATP synthase at 11 μM (6 mg ml⁻¹) and the sample incubated at 22 °C for 30 s, before 3.5 μl was placed on glow-discharged holey gold grid (Ultrafoils R1.2/1.3, 200 Mesh). Grids were blotted for 3 s at 22 °C, 100% humidity and flash-frozen in liquid ethane using a FEI Vitrobot Mark IV (total time for sample application, blotting and freezing was 45 s).

**Data collection**. Grids were transferred to a Thermo Fisher Talos Arctica transmission electron microscope (TEM) operating at 200 kV and screened for ice thickness and particle density. Grids were subsequently transferred to a Thermo Fisher Titan Krios TEM operating at 300 kV equipped with a Gatan BioQuantum energy filter and K3 Camera at the Pacific Northwest Centre for Cryo-EM at OHSU. Images were recorded automatically using serial EM at ×81,000 magnification yielding a physical pixel size of 1.08 Å (K3 operating in super resolution mode: 0.54 Å per pixel micrographs). A total dose of 48 electrons per Å² was used spread over 77 frames, with a total exposure time of 3.5 s. 9342 movie micrographs were collected (Supplementary Fig. 1).

**Data processing**. MotionCorr2[54] was used to correct local beam-induced motion and to align resulting frames, with 9 × 9 patches and binning by a factor of two. Defocus and astigmatism values were estimated using Gctf[38] and 8290 micrographs were selected after exclusion based on ice contamination, drift and astigmatism. 1361 particles were picked manually and subjected to 2D classification to generate templates for autopicking in RELION-3.0[55], yielding 1,349,270 particles. Images were then inspected manually to remove particles located in regions containing ice or aggregated protein to yield 1,111,931 particles. These particles were binned by a factor of four and subjected to 2D classification generating a final dataset of 709,190 particles. These particles when then re-extracted at full resolution and further classified into 3D classes using a low pass filtered cryo-EM model generated from a previous study[14], yielding maps related by a rotation of the central stalk (355,964, 179,005 and 174,221 particles). Focused classification, using a mask encompassing the $F_o$ motor, was implemented without performing image alignment in Relion 3.0, yielding nine defined sub-classes (five for State 1, two for State 2, and two for State 3). 3D classification parameters were iteratively modified to increase the number of defined classes observed, reaching a maximum of nine defined states. By examining the position of subunit γ in each of the sub-states, we were able to unambiguously assign the c subunits, so that their relative position could be compared between each sub-state (Supplementary Fig. 6). Refinement using a mask of the $F_o$ region was performed on Sub-state 3 A to improve clarity in the membrane region (Supplementary Fig. 10). See Supplementary Fig. 2 for detailed flowchart describing this classification and Supplementary Fig. 24 for FSC curves.

**Model building**. Models were built and refined in Coot[56], PHENIX[57] and ISO-LDE[58] using pdb IDs 3oaa[25] (*E. coli* $F_1$-ATPase), 1abv[59] (N-terminal domain of *E. coli* subunit δ) and 6n2y[17] (*Bacillus sp.* PS3 ATP synthase) as guides. Auto-sharpening implemented in PHENIX[57] was performed on the focused map of Sub-state 3A, and this higher detailed information was used to build the $F_o$ region before transferring to the other sub-states for local refinement. See Supplementary Fig. 25 for model to map FSC curves and Supplementary Table 1 for all data collection and refinement statistics.

**Mass spectrometry**. Protein complexes were digested with trypsin and dried using a vacuum concentrator (SpeedVac, ThermoFisher Sciencitific). The co-purified lipids were resuspended in 60% acetonitrile by sonication for 10 min and analyzed by LC-MS based lipidomics[24]. Briefly, the lipid samples were separated on a C18 column (Acclaim PepMap 100, C18, 75 μm × 15 cm; Thermo Scientific) by Dionex UltiMate 3000 RSLC nano System, and then analyzed by a hybrid LTQ-Orbitrap XL mass spectrometer (Thermo Scientific). A binary buffer system was used with buffer A [ACN: H2O (60:40), 10 mM ammonium formate, 0.1% formic acid] and buffer B [IPA: ACN (90:10), 10 mM ammonium formate, 0.1% formic acid]. The phospholipids were separated with a gradient of 32 to 99% buffer B at a flow rate of 300 nl min⁻¹ over 30 min. The LTQ-Orbitrap XL was operated in negative ion mode and in data-dependent acquisition with one MS scan followed by three MS/MS scans.

**Reporting summary**. Further information on research design is available in the Nature Research Reporting Summary linked to this article.

## Data availability

Maps and models have been deposited in the EMDB and PDB with the following accession codes: Sub-state 1A; EMD-20167 and PDB-6OQR [https://doi.org/10.2210/pdb6OQR/pdb]. Sub-state 1B; EMD-20168 & PDB-6OQS [https://doi.org/10.2210/pdb6OQS/pdb]. Sub-state 1C; EMD-20169 and PDB-6OQT [https://doi.org/10.2210/pdb6OQT/pdb]. Sub-state 1D; EMD-20170 and PDB-6OQU [https://doi.org/10.2210/pdb6OQU/pdb]. Sub-state 1E; EMD-20454 and PDB-6PQV [https://doi.org/10.2210/pdb6PQV/pdb]. Sub-state 2A; EMD-21854 and PDB-6WNQ [https://doi.org/10.2210/pdb6WNQ/pdb]. Sub-state 2B; EMD-20171 and PDB-6OQV [https://doi.org/10.2210/pdb6OQV/pdb]. Sub-state 3A; EMD-20172 and PDB-6OQW [https://doi.org/10.2210/pdb6OQW/pdb]. Sub-state 3B; EMD-21855 and PDB-6WNR [https://doi.org/10.2210/pdb6WNR/pdb]. Sub-state 3A $F_o$ focused: EMD-21419 and PDB-6VWK [https://doi.org/10.2210/pdb6VWK/pdb]. The source data underlying Supplementary Fig 18 are provided as a Source Data file. Other data are available from the corresponding authors upon reasonable request.

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

## Acknowledgements

We wish to thank and acknowledge Dr. Craig Yoshioka and Dr. Claudia López (Oregon Health and Sciences University (OHSU)) for data collection and processing expertize; Dr. Thomas Duncan (Department of Biochemistry and Molecular Biology, SUNY Upstate Medical University, Syracuse, NY, USA) for initial discussions on experimental design; and Sergey Tumanov (The Victor Chang Cardiac Research Institute) for help with initial lipid analysis. A.G.S. was supported by a National Health and Medical Research Council Fellowship APP1159347 and Grant APP1146403. We wish to thank and acknowledge the use of the Victor Chang Cardiac Research Institute Innovation Centre, funded by the NSW Government, and the Electron Microscope Unit at UNSW Sydney, funded in part by the NSW Government. A portion of this research was supported by NIH grant U24GM129547 and performed at the Pacific Northwest Centre for Cryo-EM at OHSU and accessed through EMSL (grid.436923.9), a DOE Office of Science User Facility sponsored by the Office of Biological and Environmental Research.

## Author contributions

A.G.S., R.M.B., and R.I. conceived the study and wrote the manuscript. A.G.S. supervised the study. M.S. performed the formal analysis of the study (purified protein, prepared cryo-EM grids, performed reconstructions). J.L.W. built and refined the atomic models into the cryo-EM maps. D.W. performed mass spectrometry based lipid analysis and C.V.R. supervised mass spectrometry analysis. Y.C.Z. performed particle picking, data analysis and preliminary lipid analysis. R.M.B. analyzed data and proposed the model of elastic coupling.

## Competing interests

The authors declare no competing interests.
