## [Peer Review File · Nature Communications]

Reviewers' comments:

Reviewer #1 (Remarks to the Author):

Sobti et al., present the structure of the E.coli ATP synthase in 9 different conformations. Five conformations in state 1, two in state 2 and two in state 3. The structures were obtained by single particle cryoEM on a cysteine free version of the native enzyme allowing future studies into the structure/function relationship of the enzyme and the correlation of existing biochemical studies with structural changes. The most interesting part of the manuscript was the five different conformations of the state 1 enzyme which allows predictions of how symmetry mismatch between the F1 catalytic domain and the Fo rotor works to effectively generate energy conversion. This observation makes the manuscript stand out from the plethora of papers on the different conformational states of the F1Fo ATP synthase which are currently flooding the field.

The main criticism I have of the manuscript is a lack of detail describing how the authors reached the conclusions described in the results section.

E.g. on line 92, the authors state that "the rotor was observed in four discrete rotational positions relative to the Fo stator: one in each of states 2 and 3 and two in state 1". My first questions would be how did the authors decide on the states? The authors provide the answer on line 172 but not in the results section. Another question would be how did they determine the four discrete rotational positions? What key observation did they observed which allowed them to make this conclusion? e.g. why do they say the C-subunits of State 1a are in a different position to state 1E? This was not described. Thus, the results need to be more detailed and describe the structure and what lead to the comments made rather than just stating a conclusion in passing. (I actually found the discussion section a better description of the results than the results section.)

Another issue is a discussion of the lipids. In which direction was the Fo turning in relation to F1 at the point of vitrification? Could the direction of rotation explain the trapped lipids? If the rotor was turning away from the a-subunit at the point where the lipids were observed, how do the lipids stay in position? These possibilities need to be discussed.

The last main issue is the assignment of the nucleotides. Based on the resolution of the different maps, how can you be certain about the different assignment of nucleotides found in the binding sites? what criteria did you use? how do you know the differences are real and not noise? How can you accurately assess the differences in density in the different binding pockets relative to differences in local resolution? What do the density look like for the residues which bind the nucleotides? Is there any difference in side chain positions for the residues involved in binding the gamma phosphate and water? Please show this information?

Other issues or specific points:

- 1) more specific information in the results section e.g. line 66: what is the "detailed information"
- 2) line 76: what are the "described movements and rotations"?
- 3) how did you identify the states?
- 4) how did you identify the sub-states?
- 5) which direction is the rotor likely to be moving in at the time of vitrification?
- 6) how did you determine the c-ring was in a different position between State 1A and State 1B?
- 7) On line 102, the authors mention the gamma subunit relative to the alpha and beta subunits was in the same rotational position in all 4 states. Which 4 states are being referred to? please indicate the position/direction of the gamma subunit in the figures.
- 8) line 108 has almost the same information in as line 62.
- 9) Line 109/110. "although their overall resolution was lower than those from the complete ensemble" what was the resolution being referred to? e.g. what was the overall resolution? what is the resolution of the ensemble? what is the ensemble that is being referred to?
- 10) line 112. Define PS3.
- 11) line 125-129 needs to be illustrated in a figure (perhaps add to figure 4)

- 12) line 148. An extra space is located after the Greek character.
- 13) line 175: how is the elastic flexibility measured?
- 14) line 198 indicates this manuscript is proposing a novel pivoting mechanism. If so, this needs to be explained in more detail. e.g. where are the pivot points located? what observation leads the authors to draw this conclusion? Why is the hinge model not appropriate? where would the hinge region be located in the E.coli structure if it was to occur.
- 15) line 208. What do the authors mean by irradiation especially in terms of chemiosmotic hypothesis?
- 16) line 222. The close proximity of residues 10-20 of the a-subunit to the N-terminus of the b-subunit needs to be shown in a figure.
- 17) line 223. Is the large interacting surface with multiple contacts shown in a figure?
- 18) line 290. Remove "the".

Figure1: please add some atomic model fitted to densities.

Figure 2: please indicate direction of gamma subunit. How do you know the two c-subunits colored in 1d are equivalent?

Figure S3: Please add state 1B,C,D and E to one figure to improve the clarity in describing how the subunits move.

Figure 3a: What are the lines originating from the center of the schematics representing? which direction is the enzyme working in? For the density model, it is probably best to show a slice through the density indicating position of the gamma subunits to the c-ring and a-subunit.

figure 3b: How was this figure generated? Who and how were the dwell times measured? What do the open boxes mean? how were the values obtained?

Figure 4: what is depicted in pink? Where is the N and C-terminus of the a-subunit? where is the N-terminal helix of the b-subunit?

Figure 5: how are figure 5a and 5b related? please add a rotation axis to the figure? Which direction would the c-ring be turning at the time of vitrification or before inhibition?

Figure 6: how can you be sure of the differences between the bound nucleotides? what are the densities like for the residues which bind to the nucleotides, specifically the residues involved in coordinating the water and third phosphate? What is the relative local resolution at the binding sites?

Please show the electron density of the binding sites. Also difference maps of the nucleotide densities at the different sites.

Figure S8: what are the bands in the SDS gel that aren't labelled?

Figure S9: please also show the FSC of the model to the data.

Reviewer #2 (Remarks to the Author):

'Cryo-EM structures provide insight into how E. coli F1Fo ATP synthase accommodates the F1/Fo symmetry mismatch' by Sobti et al. describes a structural characterisation of the E. coli ATP synthase complex, a long-studied complex and the subject of many functional and mutational experiments. Maps of several rotary states are solved to 3-4 Å resolution, which is a substantial improvement on previously published structures for this complete complex. The work appears to be methodologically sound, and the resolution of the maps should allow for a detailed atomic model to be built. Nevertheless, the new insight provided by this study is limited, and my opinion is that it does not represent a sufficient advance to merit publication in Nature Communications. Against a backdrop of existing high-resolution structures of bacterial (Guo et al eLife 2019), chloroplast (Hahn et al Science 2018) and mitochondrial (Murphy et al Science 2019; Gu et al Science 2019) F-type ATP synthases, as well as previously published lower resolution structures of other complexes including the E. coli ATP synthase (Sobti et al 2019 eLife) the atomic model appears to be entirely in line with predictions; if this is not the case, differences to the current state of knowledge should be outlined more clearly in the text. The authors demonstrate flexing of the peripheral stalk between different states of the complex, which has previously been demonstrated for bacterial (Guo et al eLife 2019), chloroplast (Hahn et al Science 2018) and

mitochondrial (Zhou et al eLife 2015) complexes. The degree of peripheral stalk flexing appears to be of similar magnitude, or perhaps somewhat less extreme, than that demonstrated previously. This work presents substates of a given rotational state which demonstrate the same relationship of the central stalk to the F1 head, but differing c-ring position (states 1A and 1E). Similar results, in which F1 rotates together with the central stalk between substates, have been shown for the bovine mitochondrial complex – movement facilitated primarily by the peripheral stalk (Zhou et al eLife 2015) and the *Polytomella* mitochondrial complex – movement facilitated primarily by flexing in the OSCP subunit (Murphy et al Science 2019). Although the observations are well made, they have not altered my thinking regarding the mechanism of flexible coupling in ATP synthase complexes, but rather confirm published results.

I commend the authors for their mention that some ~40% of particles did not form well-resolved classes, however I question their assertion that these unresolved particles ‘probably represent molecules with conformations that are between the states identified’. What evidence is available to suggest the unclassified states lie between, rather than outside, the spectrum of movement defined by the identified states? The higher-energy conformations (corresponding to a greater deformation from equilibrium) should be less populous, which one would expect would lead to them not forming well-resolved classes. Either way, the fact remains that almost half of all ATP synthase complexes imaged did not fall into the states discussed. It would therefore not be valid to discuss the absence of any states from the dataset, as for example ‘in states 2 and 3 the spring is close to its relaxed state in the observed single Fo states, and the adjacent Fo states are inaccessible because thermal excitation is insufficient to stretch the spring through the required 1/10 of a rev (Fig. 3b).’

The authors have shown several ‘tube-like densities’ which they attribute to co-purified lipids. They may well be right – we would expect that the surface of the c-ring is in contact with membrane lipids everywhere except where it directly contacts the stator, and it would indeed be very surprising if this weren’t the case. The lipids are not well enough resolved to discuss their identity or specific interactions with the protein complex, and the suggestion of ‘an intriguing mechanism whereby, in addition to direct protein-protein contacts, the strength of the stator/rotor interaction is increased by using lipids to bridge between the interacting surfaces’ is vastly overstating the significance of this finding.

The rotational state of the F1 head appears to closely resemble that reported by Cingolani & Duncan, NSMB 2011, except that the catalytic sites all contain ADP, in line with the high concentrations of ADP under which the complex was prepared. I would encourage the authors to make the densities of Figure 6b easier to see.

In summary, this paper seems to contain well-carried out work that will add to our knowledge of ATP synthase; nonetheless, my opinion is that it is not sufficiently novel to merit publication in Nature Communications.

Minor points:

-Some data seems to be missing from Extended Data Table 1

-The following sentences are unclear as written and should be revised:

-‘These long alpha helices form a parallel right-handed coiled-coil between residues 41 and 116, with the helices offset by 5½ residues with respect to one another, show a striking correlation to the arrangement that was predicted using crosslinking studies¹⁹ (Extended Data Fig. 4’

-‘when *E. coli* F1Fo ATP synthase was imaged in the same concentration and length of time with MgATP15’.

Reviewer #3 (Remarks to the Author):

Sobti and coworkers conduct a cryo-EM study on MgADP inhibited *E. coli* F-ATP synthase. From 3-

D classification of a large single-particle dataset, they initially obtain rotary states 1, 2, and 3 (as defined by the position of the central rotor relative to the stator subcomplex) as previously described by the same authors for enzyme analyzed in presence of ATP, albeit at lower resolution (ref. 14). Further classification reveals the presence of nine distinct rotary sub-states of the complex at nominal resolutions between 3.1-3.4 Å, with five sub-states of state 1, and each two sub-states for states 2 and 3. While the sub-states for states 2 and 3 show only subtle differences of the relative positions of rotor and stator, the state 1 sub-state A (called state 1') and sub-states B-E (state 1'') differ in the position of the rotor subcomplex (alpha3,beta3,gamma,delta,epsilon,c10) relative to the stator subunit a by 36deg. This leads the authors to their main conclusion, namely that the presence of states 1' and 1'' serves to accommodate the symmetry mismatch between alpha3beta3's three catalytic sites and the 10 proton carrying c subunits. The study also suggests that most of the elastic energy is stored in the peripheral b2 stalk, while the central rotor appears to be more rigid. As the authors rightly point out, the issue of which structural element in the rotary ATPase family allows storage of elastic energy has been controversial - and the data in the current manuscript add to the emerging picture that the answer to the question may depend on the type of rotary motor ATPase/synthase.

The structures here reported are the first near-atomic structures of the well characterized E. coli ATP synthase and together with the authors' overall conclusion, the study should be of general interest to those studying long-range energy coupling in rotary ATPases and related systems. However, there are several issues, some of technical nature, that preclude recommendation for publication of the manuscript in its current form.

Specific points:

(1) This study describes a dizzying array of conformational and rotational "states", "sub-states", "sub-steps" and "sub-classes". For example, the paragraph heading on page 3 announces "Rotational sub-states" and first mentions "States 1-3", then these states are divided into "sub-classes", which are then termed "States" again, then there is a c-ring "sub-step", without defining what the c-ring parent "step" is, and later on on page 5, the sub-classes are termed "sub-states". It is acknowledged that the system is able to adopt many conformational and rotations states, but it is unclear what constitutes a "State" or a "sub-state", and a "step" and a "sub-step". Is there maybe a better way - e.g. call every distinct conformation a "state"? Or consistently reserve "State" for "States 1-3" and "sub-state" for all the sub-classes?

(2) On page 7, the authors state that the preparation of the complex is in the "autoinhibited" state or conformation. However, they then model a total of nine different conformations of the enzyme and they postulate that all of these 'states' and 'sub-states' lie on the reaction coordinate representing intermediates of the catalytic cycle of the enzyme (Figure 3b). However, isn't it possible that the autoinhibited enzyme adopts one or several off-catalytic pathway conformations that are not present during active turnover? And even if sub-states 1A-E are on the reaction coordinate, how do the authors know the order of the sub-states as implied in video 1?

(3) The authors state (page 3) that MgADP was used to stabilize the complex. How was this stabilizing effect quantified or determined? Is this statement based on results from the current study, or is this MgADP stabilization a prior established observation? What do the authors mean by "the contribution made by nucleotides on the regulation of the F1 motor"? With MgATP the motor runs, with MgADP it doesn't.

(4) Please provide the specific activity in the more common $\mu\text{moles}/(\text{min} \times \text{mg})$. Is the ATPase activity of the preparation sensitive to the inhibitor DCCD? In other words, is the ATPase tightly coupled to the proton channel? Some detergents are known to uncouple the E. coli ATP synthase (e.g. see Tsunoda et al., FEBS Lett. 470, 244 (2000)).

(5) Extended Figure 2: 50% of the particles are in State 1. Based on the free energy reaction

coordinate in Figure 3b, wouldn't the authors expect that States 2 & 3 are the most populated states, as opposed to what's observed? What are the open squares in Figure 3b?

(6) In previously published maps of ATP synthases, the c-ring carboxyl and the a subunit arginine appear not to be engaged in a salt bridge (e.g. *Polytomella*, 6rd7; chloroplast, 6fkf; yeast mitochondrial, 6b2z). Do the authors have an explanation why *E. coli* F₀ is different? Could a salt bridge be a hallmark of the (epsilon/MgADP) inhibited conformation?

(7) The heading on page 7 and the accompanying legend to Figure 6 state that "MgADP induces a conformational change in the F1 motor" - however, all here described maps showed the same conformation (the "autoinhibited" state), and only one conformation is shown in the figure. If this conformation change is significant in terms of mechanism, it may be helpful to show a comparison to the previous study where ATP was present during sample preparation.

(8) Only one value for the resolution of each map is given, and from the fact that overall resolution of each state before sub-classification is better than after, it seems that alignment and classification (and resolution determination) is dominated by the ATPase (alpha3beta3gamma) subcomplex. In other words, resolution appears anisotropic. It may be helpful to report resolutions for F1 and F₀ sub-complexes separately, and/or show a color coded resolution map (e.g. ResMap).

(9) Related to point (5) - it is surprising that the map for State 1, which is composed of a mixture of sub-states (or sub-classes) A-E has a better overall resolution (2.9 Å) than the individual sub-classes (3.1-3.3 Å). How is that possible?

(10) The authors see non-proteinaceous density at the interface of the c-ring and the a subunit and they speculate that this density represents lipid molecules. Building on this speculation, they speculate further that these lipid molecules may have a functional role by increasing the interaction surface between c-ring and subunit a. This is an interesting hypothesis, but since the resolution is insufficient to confidently model lipid molecules, it would help to conduct a biochemical analysis to show the presence of lipids, e.g. by mass spectrometry or thin layer chromatography. Such an analysis, if lipid is found, would increase confidence that the assignment is correct, as it is also possible that the density is due to stably bound molecules of the detergent used to solubilize the complex (digitonin).

(11) The validation reports indicate that residue 419 of the alpha subunit was modeled as asparagine but that the primary sequence is a lysine. Did the authors analyze a mutant enzyme (besides all the Cys to Ala mutations) and if so, does the K419N mutation affect activity?

Nature Communications (Sobti et al: "Cryo-EM structures provide insight into how E. coli F₁F_o ATP synthase accommodates the F₁/F_o symmetry mismatch").

Response to reviewers' comments (reviewers' comments in blue; response in red typeface):

Reviewer #1 (Remarks to the Author):

Overview: We are most grateful for this reviewer's many helpful and constructive suggestions for improving the manuscript and have revised it along the lines suggested. We have provided additional details to enable the reader to follow precisely how we reached the conclusions described in the results section. We have also discussed in greater detail the orientation of the lipids relative to the motor and the assignment of nucleotides.

The main criticism I have of the manuscript is a lack of detail describing how the authors reached the conclusions described in the results section.

E.g. on line 92, the authors state that "the rotor was observed in four discrete rotational positions relative to the Fo stator: one in each of states 2 and 3 and two in state 1". My first questions would be how did the authors decide on the states? The authors provide the answer on line 172 but not in the results section. Another question would be how did they determine the four discrete rotational positions? What key observation did they observed which allowed them to make this conclusion? e.g. why do they say the C-subunits of State 1a are in a different position to state 1E? This was not described. Thus, the results need to be more detailed and describe the structure and what lead to the comments made rather than just stating a conclusion in passing. (I actually found the discussion section a better description of the results than the results section.)

We have clarified how we assigned each state in the main text and have added an extended data figure to explain this in greater detail:

"By examining the position of subunit γ in each of the sub-states, we were able to unambiguously assign the c subunits, so that their relative position could be compared between each sub-state (Extended Data Fig. 6)."

Another issue is a discussion of the lipids. In which direction was the Fo turning in relation to F1 at the point of vitrification? Could the direction of rotation explain the trapped lipids? If the rotor was turning away from the a-subunit at the point where the lipids were observed, how do the lipids say in position? These possibilities need to be discussed.

We have expanded the text to discuss these issues more fully.

The Fo motor should not be under load and rotating at the point of vitrification, because MgADP has been added to the sample. This has been clarified at the start of the results section:

"MgADP was used in an attempt to lock the rotor of the F1-ATPase in a single rotational position, to investigate the flexible coupling between the F1 and Fo motors, as well as the contribution made by nucleotides on the regulation of the F1 motor. Hence, the F1Fo ATP synthase imaged here should not be undergoing ATP synthesis or hydrolysis, and therefore should not be rotating under the conditions imaged in this study."

During synthesis, the c-ring would rotate away from the lipids whereas during hydrolysis the c-ring would rotate towards the lipids. Hence, it is possible that ATP released during cell lysis could be hydrolysed and cause rotation of the c-ring towards the lipid. However, because the complex was purified over a period of hours, it is likely that it is in equilibrium and was not undergoing and rotation by the time it was vitrified. This point has now been discussed in the Discussion: "During synthesis the Fo motor would rotate in a clockwise

direction when viewed from the periplasm (Fig. 5). This would result in c-ring rotation away from the lipid bridge, hence if the lipid were to remain in place under synthesis the interaction to the a subunit must be stronger than that to the c subunit.”

The last main issue is the assignment of the nucleotides. Based on the resolution of the different maps, how can you be certain about the different assignment of nucleotides found in the binding sites? what criteria did you use? how do you know the differences are real and not noise? How can you accurately assess the differences in density in the different binding pockets relative to differences in local resolution? What do the density look like for the residues which bind the nucleotides? Is there any difference in side chain positions for the residues involved in binding the gamma phosphate and water? Please show this information? We have provided an improved Figure 1 together with an additional Extended Data Figure showing difference maps. The density in the nucleotide region is very detailed, with clear 3 Å resolution features present. Hence, we were able to distinguish the difference between ATP and ADP, and changes in side chains positions (Extended Data Fig. 19).

Other issues or specific points:

1) more specific information in the results section e.g. line 66: what is the “detailed information”

The more detailed information relates to the increased resolution of the maps. We have added an additional supplementary figure (Extended Data Fig. 4) to compare State 1 and Sub-state 1C, which describes this increased detail.

2) line 76: what are the “described movements and rotations”?

We have removed these words to clarify the text. Later in the paragraph these movements are now described as: “majority of the flexibility was seen in the peripheral stalk which bends and twists, together with minor movements within the central stalk”, “torsional flexing of the entire complex” and “single c-ring sub-step in the Fo motor”.

3) how did you identify the states?

The states were filtered using the program Relion, as described in the methods. We identified the position of the rotor using the relative position of subunit gamma, which is now described in the text: “By examining the position of subunit γ in each of the sub-states, we were able to unambiguously assign the c subunits, so that their relative position could be compared between each sub-state (Extended Data Fig. 6).”

4) how did you identify the sub-states?

The methods section has been extended to further clarify this. Relion 3D classification was used to sort particles into different sub-states. The position of subunit gamma was used to unambiguously assign the rotary position of the rotor.

5) which direction is the rotor likely to be moving in at the time of vitrification?

As stated above, and now in the text, the enzyme should not be rotating at the time of vitrification as a consequence of the high levels of MgADP present: “MgADP was used in an attempt to lock the rotor of the F₁-ATPase in a single rotational position, to investigate the flexible coupling between the F₁ and F₀ motors, as well as the contribution made by nucleotides on the regulation of the F₁ motor. Hence, the F₁F₀ ATP synthase imaged here should not be undergoing ATP synthesis or hydrolysis, and therefore should not be rotating under the conditions imaged in this study.”

6) how did you determine the c-ring was in a different position between State 1A and State 1B?

As discussed in point 3 and above, we used the position of subunit gamma to assign the rotational position of the c-ring: “By examining the position of subunit γ in each of the sub-states, we were able to unambiguously assign the c subunits, so that their relative position could be compared between each sub-state (Extended Data Fig. 6).”

7) On line 102, the authors mention the gamma subunit relative to the alpha and beta subunits was in the same rotational position in all 4 states. Which 4 states are being referred to? please indicate the position/direction of the gamma subunit in the figures.

We are referring all the structures presented in this study. This has been clarified in the text along with an additional supplementary figure has been added to aid this explanation: “Furthermore, the position of the γ subunit relative to the α and β subunits was in the same rotational position in all sub-states (Extended Data Fig. 9), although rotated relative to the peripheral stalk, with the F_1 enzyme locked in the same rotational state across all structures observed in this study.”

8) line 108 has almost the same information in as line 62.

Line 108 referred to the classified sub-states, whereas line 62 referred to just the three main states. The text has been changed to clarify this along with two additional supplementary figures (Extended Data Fig. 4 and 5) to showcase this detailed information and compare to the unclassified data.

9) Line 109/110. “although their overall resolution was lower than those from the complete ensemble” what was the resolution being referred to? e.g. what was the overall resolution? what is the resolution of the ensemble? what is the ensemble that is being referred to?

The resolution being compared is that between States and the Sub-states. This has been clarified in the text to discuss how the F_1 -ATPase is dominating the overall FSC calculation: “Although these reconstructions gave the highest numerical resolution, the Fourier Shell Correlation (FSC) appeared to be dominated by the F_1 -ATPase, with information in the F_0 region blurred significantly (Extended Data Fig. 3 and 4). Hence, masked classification focused on the F_0 stator was used on each of the main states to reveal sub-states describing movements of the complex along with increased detail in the F_0 region (Extended Data Fig. 2, 3, 4 and 5).”

10) line 112. Define PS3.

Bacillus sp. PS3 has been added to the text and Figure legends.

11) line 125-129 needs to be illustrated in a figure (perhaps add to figure 4)

An additional supplementary figure (Extended Data Fig. 15) has been added to describe this text.

12) line 148. An extra space is located after the Greek character.

This space has been removed.

13) line 175: how is the elastic flexibility measured?

We have not measured the elastic modulus of this component, but merely refer to the potential elastic nature of it. The text has been modified to make this clear.

14) line 198 indicates this manuscript is proposing a novel pivoting mechanism. If so, this needs to be explained in more detail. e.g. where are the pivot points located? what observation leads the authors to draw this conclusion? Why is the hinge model not appropriate? where would the hinge region be located in the E.coli structure if it was to occur.

To avoid confusion, the text has been modified to remove the term “pivoting”: “An alternative hypothesis is that the two mechanisms; twisting and flexing in the peripheral and central stalks (seen in this study on *E. coli*), and a hinge in subunit δ /*OSCP* (seen in *Polytomella*), could combine to allow a fluid motion across a ~ 60 - 80° sub-step of the enzyme.”

15) line 208. What do the authors mean by irradiation especially in terms of chemiosmotic hypothesis?

This has been clarified in the text. In cryo-EM, carboxylates are more prone to radiation damage. The proton translocating carboxylate in *E. coli* ATP synthase is an aspartic acid, whereas in other studied organisms it is a glutamic acid. Aspartic acid has a shorter sidechain than glutamic acid, and therefore has a reduced number of rotamer positions. Hence, after radiation damage has occurred, it is easier to hypothesize the position of the remaining atoms due to the reduced number of rotamer possibilities with the shorter sidechain.

“Radiation damage in cryo-EM causes the preferential loss of carboxylates, making the confident assignment and precise building of glutamates and aspartates difficult. Other studies of ATP synthases have focused on organisms that utilize a glutamic acid to translocate protons, whereas *E. coli* F_1F_0 ATP synthase utilizes aspartic acid, *cAsp61*, to translocate protons. As aspartic acid contains a shorter sidechain than the glutamic acid, the possible rotamer positions after electron irradiation are fewer giving an advantage over other systems when attempting to assign the conformation of this sidechain. Although we cannot unequivocally assign the precise nature of this interaction, the *cAsp61* adjacent to *aArg210* appears to be in an outward facing conformation compared to *cAsp61* residues exposed to the membrane (Fig. 5c), which corroborates crystallographic studies on the isolated *c*-ring.”

16) line 222. The close proximity of residues 10-20 of the a-subunit to the N-terminus of the b-subunit needs to be shown in a figure.

An additional supplementary figure (Extended Data Fig. 15) has been added to describe this text.

17) line 223. Is the large interacting surface with multiple contacts shown in a figure?

An additional supplementary figure (Extended Data Fig. 15) has been added to describe this text.

18) line 290. Remove “the”.

We have removed the extra “the” from the text.

Figure 1: please add some atomic model fitted to densities.

The figure has been updated to include fitted atomic models.

Figure 2: please indicate direction of gamma subunit. How do you know the two c-subunits colored in 1d are equivalent?

The direction of the gamma subunit was already included in the original figure (Fig. 2 b). As explained previously, the c subunits were assigned based on the position of the gamma subunit.

Figure S3: Please add state 1B,C,D and E to one figure to improve the clarity in describing how the subunits move.

We have added a new figure to show all the sub-states in one figure (Extended Data Fig. 8)

Figure 3a: What are the lines originating from the center of the schematics representing? which direction is the enzyme working in? For the density model, it is probably best to show a slice through the density indicating position of the gamma subunits to the c-ring and a-subunit.

The lines are now explained in the figure legend. The density model was used to describe where the information came from and we have attempted to show the density model as a slice through the rotor as requested, however the figure became too confusing. With the additional supplementary figure (Extended Data Fig. 6) which shows the assignment of the c-ring position.

figure 3b: How was this figure generated? Who and how were the dwell times measured? What do the open boxes mean? how were the values obtained?

The figure was generated using Adobe Illustrator and is merely a rough schematic of a linear spring. We have updated the figure legend to improve the clarity of the figure.

Figure 4: what is depicted in pink? Where is the N and C-terminus of the a-subunit? where is the N-terminal helix of the b-subunit?

The b subunits were shown in pink, the figure legend has been updated to clarify this: “(a) Sub-state 3A F₀ focused cryo-EM map of the *E. coli* F₀ stator, masked to remove c-ring for clarity (subunit a in orange, subunit b in pink and lipid-like densities in wheat).”

Figure 5: how are figure 5a and 5b related? please add a rotation axis to the figure? Which direction would the c-ring be turning at the time of vitrification or before inhibition?

We have added a rotation axis (between 5a and 5b) and the direction of rotation for synthesis to the figure. The enzyme should not be rotating at the time of vitrification. Before inhibition, the enzyme is likely to be stationary, due to the lack of nucleotide in buffers. However, at the time of lysis, the enzyme may be undergoing ATP hydrolysis.

Figure 6: how can you be sure of the differences between the bound nucleotides? what are the densities like for the residues which bind to the nucleotides, specifically the residues involved in coordinating the water and third phosphate? What is the relative local resolution at the binding sites?

Please show the electron density of the binding sites. Also difference maps of the nucleotide densities at the different sites.

This point has been clarified earlier in the rebuttal (the third major criticism). Difference maps of the nucleotides have been included in a supplementary figure along with density for the surrounding residues (Extended Data Figure 19). The local resolution (as calculated in Relion) at the binding sites is 2.75-3.25 Å across the maps (see Extended Data Figure 3)

Figure S8: what are the bands in the SDS gel that aren't labelled?

We performed mass spec fingerprinting on the contamination bands seen in this SDS PAGE gel, and these were identified as Ribonuclease E GroEL and ElaB from *E. coli*.

FigureS9: please also show the FSC of the model to the data.

These have been added as a new supplementary figure (Extended Data Fig. 24)

Reviewer #2 (Remarks to the Author):

‘Cryo-EM structures provide insight into how *E. coli* F1Fo ATP synthase accommodates the F1/Fo symmetry mismatch’ by Sobti et al. describes a structural characterisation of the *E. coli* ATP synthase complex, a long-studied complex and the subject of many functional and mutational experiments. Maps of several rotary states are solved to 3-4 Å resolution, which is a substantial improvement on previously published structures for this complete complex. The work appears to be methodologically sound, and the resolution of the maps should allow for a detailed atomic model to be built. Nevertheless, the new insight provided by this study is limited, and my opinion is that it does not represent a sufficient advance to merit publication in *Nature Communications*. Against a backdrop of existing high-resolution structures of bacterial (Guo et al eLife 2019), chloroplast (Hahn et al Science 2018) and mitochondrial (Murphy et al Science 2019; Gu et al Science 2019) F-type ATP synthases, as well as previously published lower resolution structures of other complexes including the *E. coli* ATP synthase (Sobti et al 2019 eLife) the atomic model appears to be entirely in line with predictions; if this is not the case, differences to the current state of knowledge should be outlined more clearly in the text. The authors demonstrate flexing of the peripheral stalk between different states of the complex, which has previously been demonstrated for bacterial (Guo et al eLife 2019), chloroplast (Hahn et al Science 2018) and mitochondrial (Zhou et al eLife 2015) complexes. The degree of peripheral stalk flexing appears to be of similar magnitude, or perhaps somewhat less extreme, than that demonstrated previously. This work presents substates of a given rotational state which demonstrate the same relationship of the central stalk to the F1 head, but differing c-ring position (states 1A and 1E). Similar results, in which F1 rotates together with the central stalk between substates, have been shown for the bovine mitochondrial complex – movement facilitated primarily by the peripheral stalk (Zhou et al eLife 2015) and the *Polytomella* mitochondrial complex – movement facilitated primarily by flexing in the OSCP subunit (Murphy et al Science 2019). Although the observations are well made, they have not altered my thinking regarding the mechanism of flexible coupling in ATP synthase complexes, but rather confirm published results.

We thank the reviewer for confirming the quality of the work. However, we do feel that the new information obtained is substantial and represents a considerable advance in the field, especially in the context of the additional information now provided regarding the presence of lipids. Although it was clear from previous studies that there needed to be some coupling, this is the first detailed analysis that shows how it is achieved in *E. coli*, where indeed it is different to the other systems that have been described. Because *E. coli* is the organism that has been used extensively for a wealth of genetic and biochemical investigations, the results obtained will be of broad general interest. We have added further discussion along with supplementary figures to describe the location and implications of these essential residues. In addition, this is the first study to identify the presence of such lipids in the distal region of the stator/rotor interface and raises the possibility of their making an unanticipated contribution to the interactions.

I commend the authors for their mention that some ~40% of particles did not form well-resolved classes, however I question their assertion that these unresolved particles ‘probably represent molecules with conformations that are between the states identified’. What evidence is available to suggest the unclassified states lie between, rather than outside, the spectrum of movement defined by the identified states?

~60% of particles at the 3D classification stage is in-line with other studies, and much higher than studies on other flexible molecules (e.g. Nguyen et al. 2018 doi: 10.1038/s41586-018-

0062-x). It is our hypothesis that the unresolved particles probably represent molecules with conformations that are near the sub-states identified. This assumption is now clearly stated in the text, and we have clarified this by changing the term from “between” to “around”, and added the possibility that we cannot rule out conformations well beyond those identified. The sorting of particles is likely due to the limitations of the algorithm implemented in Relion, filtering sub-states which are separated by a significant degree and incorporating particles close by. This would leave small numbers of particles in the “gaps” between sub-states that do not refine to meaningful reconstructions. The text has been modified accordingly: “We were unable to confidently classify the remaining ~40% of the particles into discrete rotational sub-states due to the weak peripheral stalk density in maps generated with them. It is our assumption that these unclassified particles represent molecules with conformations that are around the states identified and, because of their torsional variability, have insufficient signal for the sorting algorithm to succeed in classifying them. Though we cannot rule out that these particles represent conformations well beyond those identified in this study.”

The higher-energy conformations (corresponding to a greater deformation from equilibrium) should be less populous, which one would expect would lead to them not forming well-resolved classes. Either way, the fact remains that almost half of all ATP synthase complexes imaged did not fall into the states discussed. It would therefore not be valid to discuss the absence of any states from the dataset, as for example ‘in states 2 and 3 the spring is close to its relaxed state in the observed single Fo states, and the adjacent Fo states are inaccessible because thermal excitation is insufficient to stretch the spring through the required 1/10 of a rev (Fig. 3b).’

It is likely that a large number of particles have been frozen during their transition between the sub-states. Because these will represent a broad spectrum of different positions they will tend to be blurred out and so would not be amenable to the classification methods employed here. However, the absence of specific sub-states is still suggestive. Although our suggestions here are indeed speculation and further work will be needed to evaluate these hypotheses we think it is important to raise these possibilities and feel they will be of considerable interest to other workers in the field and will serve to stimulate further work in this area. In response to this criticism, we have modified the text to make clearer the speculative nature of our hypotheses surrounding the use of particles distributions and that further work will be needed to evaluate them.

The authors have shown several ‘tube-like densities’ which they attribute to co-purified lipids. They may well be right – we would expect that the surface of the c-ring is in contact with membrane lipids everywhere except where it directly contacts the stator, and it would indeed be very surprising if this weren’t the case. The lipids are not well enough resolved to discuss their identity or specific interactions with the protein complex, and the suggestion of ‘an intriguing mechanism whereby, in addition to direct protein-protein contacts, the strength of the stator/rotor interaction is increased by using lipids to bridge between the interacting surfaces’ is vastly overstating the significance of this finding.

We have now used focused refinement to generate better maps that show increased detail in the F_o region (Extended Data Fig. 10). This analysis shows higher detailed information for the lipids, and strengthens weak density corresponding to lipids around the c-ring. The density corresponding to the lipid bridge in these new maps is as strong as the main chain atoms, further suggesting that this material is ordered. In response to reviewer 3, we have also used mass spectrometry to demonstrate the presence of lipids in the samples used for cryo-EM. We do not think it is overstating the importance of this finding to say that it raises

the possibility of an intriguing mechanism by which lipids could contribute to the stator-rotor interaction and in the text we stress that this is a hypothesis that will require further work to evaluate.

The rotational state of the F1 head appears to closely resemble that reported by Cingolani & Duncan, NSMB 2011, except that the catalytic sites all contain ADP, in line with the high concentrations of ADP under which the complex was prepared. I would encourage the authors to make the densities of Figure 6b easier to see.

We have improved this figure and included a supplementary figure (Extended Data Fig. 19) to make it easier to appreciate how effectively the nucleotide can be identified.

In summary, this paper seems to contain well-carried out work that will add to our knowledge of ATP synthase; nonetheless, my opinion is that it is not sufficiently novel to merit publication in Nature Communications.

Minor points:

-Some data seems to be missing from Extended Data Table 1

We left this data out as felt it was misleading to include such information as “Map resolution range”. We have included the information for all other sections.

-The following sentences are unclear as written and should be revised:

-‘These long alpha helices form a parallel right-handed coiled-coil between residues 41 and 116, with the helices offset by $5\frac{1}{2}$ residues with respect to one another, show a striking correlation to the arrangement that was predicted using crosslinking studies¹⁹ (Extended Data Fig. 4’

This text has been revised to improve clarity: “The peripheral stalk of *E. coli* ATP synthase is constructed by a homodimer containing two *b* subunits. Each subunit consists of a long alpha helix with three sections (Extended Data Fig. 13). The N-terminal section (*b*1-45) resides in the membrane and braces against the *a* subunit, the middle section (*b*46-*b*135) forms a ~130 Å long right-handed coiled coil and the C-terminal section (*b*136-*b*154) which loops back to cap the attachment to the F₁-ATPase. The helices within the parallel right-handed coiled coil are offset by $5\frac{1}{2}$ residues with respect to one another. This offset shows a striking correlation to the arrangement that was predicted previously using crosslinking studies, showing a staggered homodimeric right-handed coiled coil (Extended Data Fig. 14).”

-‘when *E. coli* F1Fo ATP synthase was imaged in the same concentration and length of time with MgATP¹⁵’.

This text has been removed.

Reviewer #3 (Remarks to the Author):

Overview: We are most grateful for this reviewer's many helpful and constructive suggestions for improving the manuscript and have revised it along the lines suggested. We have modified the text to increase clarity and performed mass spectrometry to show the presence of lipids in the sample.

Specific points:

(1) This study describes a dizzying array of conformational and rotational "states", "sub-states", "sub-steps" and "sub-classes". For example, the paragraph heading on page 3 announces "Rotational sub-states" and first mentions "States 1-3", then these states are divided into "sub-classes", which are then termed "States" again, then there is a c-ring "sub-step", without defining what the c-ring parent "step" is, and later on on page 5, the sub-classes are termed "sub-states". It is acknowledged that the system is able to adopt many conformational and rotations states, but it is unclear what constitutes a "State" or a "sub-state", and a "step" and a "sub-step". Is there maybe a better way - e.g. call every distinct conformation a "state"? Or consistently reserve "State" for "States 1-3" and "sub-state" for all the sub-classes?

We thank this reviewer for their extremely helpful suggestion. We have modified the text to consistently use *State* and *Sub-state* throughout the text.

(2) On page 7, the authors state that the preparation of the complex is in the "autoinhibited" state or conformation. However, they then model a total of nine different conformations of the enzyme and they postulate that all of these 'states' and 'sub-states' lie on the reaction coordinate representing intermediates of the catalytic cycle of the enzyme (Figure 3b). However, isn't it possible that the autoinhibited enzyme adopts one or several off-catalytic pathway conformations that are not present during active turnover?

We have amended the text to discuss the possibility that the some of the sub-states seen in the autoinhibited enzyme could differ from those sampled as it passes through during its active cycle.

"The data presented in this manuscript shows the enzyme in its auto-inhibited conformation, with the F₁-ATPase in a single rotational state. Hence, further work will need to be performed on this enzyme to investigate whether the peripheral and central stalks retain the same properties when the enzyme is under load either synthesizing or hydrolyzing ATP"

Although further work, that is outside the scope of the present manuscript, will be required to resolve this question, the sub-states we have identified serve as an important starting point to stimulate further work in this area.

And even if sub-states 1A-E are on the reaction coordinate, how do the authors know the order of the sub-states as implied in video 1?

We thank the reviewer for highlighting how this video could be misinterpreted. Video 1 is not meant to imply the order of sub-states that the enzyme passes through. The video simply cycles through the sub-states to highlight the structural changes between them. We have now supplied a video legend in the revised manuscript to clarify this point.

"Video 1: Morphing between Sub-states 1A-E highlights the structural differences between the classified sub-states. The order of the morphs does not represent the direction of movement and was chosen to highlight differences not describe a progressive movement. (a) side view, (b) rotated 90 degrees relative to a, (c) viewed from the cytoplasm and (d) viewed from the periplasm. Green dot on left to indicate the transition between each sub-state."

(3) The authors state (page 3) that MgADP was used to stabilize the complex. How was this stabilizing effect quantified or determined? Is this statement based on results from the current study, or is this MgADP stabilization a prior established observation? What do the authors mean by “the contribution made by nucleotides on the regulation of the F₁ motor”? With MgATP the motor runs, with MgADP it doesn't.

We used MgADP on the basis that it would not be hydrolyzed and instead lock the F₁-ATPase to aid in sub-state classification. The text has been expanded to clarify this point: “MgADP was used in an attempt to lock the rotor of the F₁-ATPase in a single rotational position, to investigate the flexible coupling between the F₁ and F_o motors, as well as the contribution made by nucleotides on the regulation of the F₁ motor. Hence, the F₁F_o ATP synthase imaged here should not be undergoing ATP synthesis or hydrolysis, and therefore should not be rotating under the conditions imaged in this study.”

(4) Please provide the specific activity in the more common $\mu\text{moles}/(\text{min} \times \text{mg})$. Is the ATPase activity of the preparation sensitive to the inhibitor DCCD? In other words, is the ATPase tightly coupled to the proton channel? Some detergents are known to uncouple the *E. coli* ATP synthase (e.g. see Tsunoda et al., FEBS Lett. 470, 244 (2000)).

We have shown previously that this sample is sensitive to DCCD inhibition (see Sobti et. al 2016 doi: 10.7554/eLife.21598) and have added a note to this effect to the text. The units of ATP per second were used for to facilitate comparison to Ishmukhametov et. al 2010 (doi: 10.1038/emboj.2010.259), but we have now also included $\sim 9 \mu\text{moles}/(\text{min} \times \text{mg})$ as requested by this reviewer. Digitonin was chosen to solubilize the material because it was the only detergent screened that kept the enzyme coupled during size exclusion chromatography.

(5) Extended Figure 2: 50% of the particles are in State 1. Based on the free energy reaction coordinate in Figure 3b, wouldn't the authors expect that States 2 & 3 are the most populated states, as opposed to what's observed? What are the open squares in Figure 3b?

The open squares show potential states that were not observed in this study, this is now addressed in the figure legend. The free energy diagram shown in Fig. 3b is just a simple schematic and shows a possible explanation of why we observe two states (the F₁ dwell is located between a F_o dwell).

(6) In previously published maps of ATP synthases, the c-ring carboxyl and the a subunit arginine appear not to be engaged in a salt bridge (e.g. *Polytomella*, 6rd7; chloroplast, 6kfk; yeast mitochondrial, 6b2z). Do the authors have an explanation why *E. coli* F_o is different? Could a salt bridge be a hallmark of the (epsilon/MgADP) inhibited conformation?

Indeed, the potential salt bridge could be a hallmark of the epsilon/MgADP. However, further work will be required to distinguish between this possibility and whether the salt bridge is instead a feature of the uninhibited state. We have modified the text to explore this possibility: “Unlike other structural studies on related ATP synthases, the interface between the *a* and *c* subunits remained largely similar across the classified maps. This could be a hallmark of the ϵ CTD/MgADP inhibited state, a specific feature of *E. coli* F₁F_o ATP synthase or an artefact or the methods used to classify the particles. Further studies on uninhibited enzymes and ATP synthases from other organisms will be needed to delineate this structural feature.”

(7) The heading on page 7 and the accompanying legend to Figure 6 state that “MgADP induces a conformational change in the F₁ motor” - however, all here described maps showed the same conformation (the “autoinhibited” state), and only one conformation is shown in the

figure. If this conformation change is significant in terms of mechanism, it may be helpful to show a comparison to the previous study where ATP was present during sample preparation. We have added a new supplementary figure (Supplementary Figure 18) and an arrow to Figure 6 to clarify the differences between the F₁-ATPase across the solved cryo-EM maps and crystal structure.

(8) Only one value for the resolution of each map is given, and from the fact that overall resolution of each state before sub-classification is better than after, it seems that alignment and classification (and resolution determination) is dominated by the ATPase (alpha3beta3gamma) subcomplex. In other words, resolution appears anisotropic. It may be helpful to report resolutions for F₁ and F₀ sub-complexes separately, and/or show a color coded resolution map (e.g. ResMap).

We have performed local resolution implemented in Relion for all maps included in this study and presented them in an additional supplementary figure (Extended Data Fig. 3). However, this analysis was not particularly informative and visual inspection of the maps is far more insightful (Extended Data Fig. 4). We have also performed focused refinement on the F_o region of Sub-state 3A to reveal the best detail in this region (Extended Data Figure 10). This map is now used for all the figures describing the F_o region.

(9) Related to point (5) - it is surprising that the map for State 1, which is composed of a mixture of sub-states (or sub-classes) A-E has a better overall resolution (2.9 Å) than the individual sub-classes (3.1-3.3 Å). How is that possible?

As discussed above in our response to reviewer #1 (specific point 1), the F₁-ATPase dominates the FSC and so results in a lower overall resolution number when a mixture of sub-states is used. Classification reduces the overall resolution number, but provides increased detail in the F_o region. This is now discussed in greater detail in the main text.

(10) The authors see non-proteinaceous density at the interface of the c-ring and the a subunit and they speculate that this density represents lipid molecules. Building on this speculation, they speculate further that these lipid molecules may have a functional role by increasing the interaction surface between c-ring and subunit a. This is an interesting hypothesis, but since the resolution is insufficient to confidently model lipid molecules, it would help to conduct a biochemical analysis to show the presence of lipids, e.g. by mass spectrometry or thin layer chromatography. Such an analysis, if lipid is found, would increase confidence that the assignment is correct, as it is also possible that the density is due to stably bound molecules of the detergent used to solubilize the complex (digitonin).

We have performed mass spectrometry to confirm that lipids were present in the detergent-solubilized material imaged in this study. Phosphatidylethanolamines, phosphatidylglycerols and cardiolipins were all observed to be present, with cardiolipin showing an enrichment relative to the starting material. An additional supplementary figure has been included and the text has been modified: “To investigate whether the densities observed could be attributed to lipids, LC-MS based lipidomics was performed on the same detergent solubilized *E. coli* F₁F_o ATP synthase imaged in this study. Phosphatidylethanolamines, phosphatidylglycerols and cardiolipins were all observed, with an increase in the relative abundance of cardiolipins relative to *E. coli* membrane (Extended Data Fig. 17), showing that *E. coli* lipids were co-purified with the protein. Given the shape of the lipid-like densities observed along with the lipids observed by mass-spectrometry, it is highly likely that these densities represent lipids rather than detergent.”

(11) The validation reports indicate that residue 419 of the alpha subunit was modeled as

asparagine but that the primary sequence is a lysine. Did the authors analyze a mutant enzyme (besides all the Cys to Ala mutations) and if so, does the K419N mutation affect activity?

This was a mistake introduced during the building process, an oversight likely due to the large number of residues being built and modified. This has been fixed and the models have been replaced in the pdb.

REVIEWERS' COMMENTS:

Reviewer #1 (Remarks to the Author):

The authors have answered all my concerns satisfactory except one. For figure 3b I asked how the figure was made. The authors quite correctly told me which program they used to generate the image. What I wanted to know was where did the data come from which forms the bases of the figure. e.g. how was it measured? How are the values for E determined/estimated.

I would also advise the authors to carefully read through their manuscript before final submission. There was quite a number of typos and interesting sentences. Some examples include:
page 19, last paragraph: "3D classification parameters were iteratively modified in to increase the number of defined classes observed, researching a maximum of nine states"
Page 27-28. "Three lipid-like densities on the periplasmic side membrane are the strongest of densities, suggested them to be well ordered".
Supplementary figure 18A, cyro should be cryo.

Reviewer #3 (Remarks to the Author):

This is a revised manuscript. The authors have responded well to the questions and criticisms from the first round of review, and thanks to the addition of new data (lipid mass spectrometry, additional map refinement) and the expansion and editing of the text, the manuscript is greatly improved. However, before final acceptance could be recommended, a few issues related to the new data and new or revised figures need to be addressed.

- Figure 4: Main text (line 265) and the legend refers to panel (c), but there is no panel (c) in the figure. Also, line 274 refers to Figure 5c, which probably should be Figure 4c. Moreover, the distance between Asp61 and Arg210 in the original version of the manuscript was given as 2.5 and 3 Å, with 2.5 Å representing a minor clash. In the revised version, the distance is now given as 2.2 and 3.4 Å, with 2.2 Å representing a severe clash. How confident are the authors that the side chains are modeled correctly, given the still limited resolution of ~3 Å and the large number of possible rotamers for arginine residues? At this resolution, a severe clash is most often associated with a modeling error. The severe clash is especially worrisome given the authors statement that the shortness of the aspartic acid side chain represents an "advantage" when modeling its side chain conformation due to the small number of possible rotamers (page 12, line 270).

- Figure 4 and supplementary Fig. 11: How was the modeled water molecule validated? Based on high resolution crystal structure analysis, the ideal distance of ordered water molecules to H-bonding polar atoms is ~2.9 Å (see e.g. Nakasako, M. (2004) Phil. Trans. R. Soc. Lond. B (2004) 359, 1191). Please give the distances of this putative water molecule to surrounding polar atoms.

- Please state (in the legend or on the figure) which of the 8 models is shown in supplementary figures 11, 13, 15, 16.

- Fo subunits should be italics - please fix for 'a' and 'c', e.g. lines 183, 221, 285, 327, 441, 660, 665, 669 (main text). Also true for E. coli, e.g. line 189.

- Supplementary Fig. 24 is not cited in the text

Point-by-point response to reviewer comments

Reviewer #1 (Remarks to the Author):

The authors have answered all my concerns satisfactory except one. For figure 3b I asked how the figure was made. The authors quite correctly told me which program they used to generate the image. What I wanted to know was where did the data come from which forms the bases of the figure. e.g. how was it measured? How are the values for E determined/estimated.

The figure assumes that in each F_1 state, elasticity in the peripheral stalk and the rest of the structure approximates to a linear (Hookean) spring with respect to the rotation within F_0 . The parabolic curves represent the energy stored in this spring in each F_1 state, as a function of the rotation angle of F_0 : $E = \frac{1}{2} (\kappa) x^2$, where κ is the spring constant and x the displacement of the spring from its minimum. The minima for the 3 F_1 states were assumed to be equally spaced, 120° apart, and the stiffness of the spring was assumed to be the same in all 3 states. The angles of these minima relative to the 10 F_0 dwells (vertical lines) were adjusted by hand to set the Energy of Sub-state 1'' lower than that of Sub-state 1' (as required if the 3-fold higher count of particles classified as Sub-state 1'' relative to Sub-state 1' is representative of the equilibrium occupancy of these states), while at the same time setting the energies of Sub-states 2 and 3 close to their minima.

We have added further explanation to the figure legend as follows: "The parabolic curves represent the energy stored in this spring in each F_1 state, as a function of the rotation angle of F_0 : $E = \frac{1}{2} kx^2$, where k is the spring constant and x the displacement of the spring from its minimum. The minima for the three F_1 states were assumed to be equally spaced, 120° apart, and the stiffness of the spring was assumed to be the same in all three states. The angles of these minima relative to the 10 F_0 dwells (vertical lines) were adjusted by hand to set the Energy of Sub-state 1'' lower than that of Sub-state 1' (as required if the 3-fold higher count of particles classified as Sub-state 1'' relative to Sub-state 1' is representative of the equilibrium occupancy of these states), while at the same time setting the energies of Sub-states 2 and 3 close to their minima."

I would also advise the authors to carefully read through their manuscript before final submission. There was quite a number of typos and interesting sentences. some examples include:

page 19, last paragraph: "3D classification parameters were iteratively modified in to increase the number of defined classes observed, researching a maximum of nine states"

Page27-28. "Three lipid-like densities on the periplasmic side membrane are the strongest of densities, suggested them to be well ordered".

Supplementary figure 18A, cyro should be cryo.

We have carefully read the manuscript and amended as necessary. We thank the reviewer for bringing these mistakes to our attention.

Reviewer #3 (Remarks to the Author):

This is a revised manuscript. The authors have responded well to the questions and criticisms from the first round of review, and thanks to the addition of new data (lipid mass spectrometry, additional map refinement) and the expansion and editing of the text, the manuscript is greatly improved. However, before final acceptance could be recommended, a few issues related to the new data and new or revised figures need to be addressed.

- Figure 4: Main text (line 265) and the legend refers to panel (c), but there is no panel (c) in the figure.

We have amended the manuscript to read "Figure 4b"

Also, line 274 refers to Figure 5c, which probably should be Figure 4c.

We have amended the manuscript to read "Figure 4b"

Moreover, the distance between Asp61 and Arg210 in the original version of the manuscript was given as 2.5 and 3 Å, with 2.5 Å representing a minor clash. In the revised version, the distance is now given as 2.2 and 3.4 Å, with 2.2 Å representing a severe clash. How confident are the authors that the side chains are modeled correctly, given the still limited resolution of ~3 Å and the large number of possible rotamers for arginine residues? At this resolution, a severe clash is most often associated with a modeling error. The severe clash is especially worrisome given the authors statement that the shortness of the aspartic acid side chain represents an "advantage" when modeling its side chain conformation due to the small number of possible rotamers (page 12, line 270).

We have rebuilt this section using ISOLDE to improve the modelling. This strategy decreased the clashes and the model has an improved fit for the density, with distances of 2.6 and 3.3 Å. The distances are more consistent with an interaction and we thank the reviewer for their input. Although arginine residues have a large number of possible rotamers, the density in this region is very detailed (see Figure 4b right panel) and hence we are confident of the rotamer assigned.

- Figure 4 and supplementary Fig. 11: How was the modeled water molecule validated? Based on high resolution crystal structure analysis, the ideal distance of ordered water molecules to H-bonding polar atoms is ~2.9 Å (see e.g. Nakasako, M. (2004) Phil. Trans. R. Soc. Lond. B (2004) 359, 1191). Please give the distances of this putative water molecule to surrounding polar atoms.

We have included distances to adjacent polar atoms in a new figure (Supplementary Figure 12) as well as including a sphere to represent the size of the cavity in which it resides. As the reviewer points out, these bond lengths are too long for a simple H-bonding pair to a water. Hence, we have changed the text to speculate on what this density could represent. The water ion was not included in the submitted pdb and we agree that assignment of this density is speculative at ~3 Å resolution. Further work will be needed to investigate the identity of this density.

We have amended the text and figures as follows:

The figures now read "ion/water" density rather than "water".

The text reads: “As the distances to neighboring polar atoms are too large to hydrogen bond and the cavity in which this density resides has a van der Waals radius $\sim 3 \text{ \AA}$ (Supplementary Figure 12), the identity of this density is unlikely to be a single water molecule surrounded by vacuum. Instead this region could correspond to a cluster of water molecules or a hydrated ion such as Magnesium, Sodium or Phosphate. However, assigning such a density at $\sim 3 \text{ \AA}$ resolution is challenging and clearly further work would be required to evaluate the identity and importance of this region. Hence the deposited co-ordinates do not contain any atoms in this space.”

- Please state (in the legend or on the figure) which of the 8 models is shown in supplementary figures 11, 13, 15, 16.

We have amended the figure legends to include the maps/models used to make these figures.

- Fo subunits should be italics - please fix for ‘a’ and ‘c’, e.g. lines 183, 221, 285, 327, 441, 660, 665, 669 (main text). Also true for E. coli, e.g. line 189.

In line with the request from the editor, we have removed all italicised text used for subunits names.

- Supplementary Fig. 24 is not cited in the text

We have amended the text to cite Supplementary Figure 24 (now Supplementary Figure 25).